# Disentangled Cross-Modal Representation Learning with Enhanced Mutual Supervision

**Lu Gao**[*]
School of Computer Science and Engineering
Central South University
Changsha, China
244711035@csu.edu.cn

**Wenlan Chen**[*]
School of Computer Science and Engineering
Central South University
Changsha, China
244701041@csu.edu.cn

**Daoyuan Wang**
School of Computer Science and Engineering
Central South University
Changsha, China
244701040@csu.edu.cn

**Fei Guo**[†]
School of Computer Science and Engineering
Central South University
Changsha, China
guofei@csu.edu.cn

**Cheng Liang**[†]
School of Information Science and Engineering
Shandong Normal University
Jinan, China
alcs417@sdnu.edu.cn

## Abstract

Cross-modal representation learning aims to extract semantically aligned representations from heterogeneous modalities such as images and text. Existing multimodal VAE-based models often suffer from limited capability to align heterogeneous modalities or lack sufficient structural constraints to clearly separate the modality-specific and shared factors. In this work, we propose a novel framework, termed **D**isentangled **C**ross-**M**odal Representation Learning with **E**nhanced **M**utual Supervision (DCMEM). Specifically, our model disentangles the common and distinct information across modalities and regularizes the shared representation learned from each modality in a mutually supervised manner. Moreover, we incorporate the information bottleneck principle into our model to ensure that the shared and modality-specific factors encode exclusive yet complementary information. Notably, our model is designed to be trainable on both complete and partial multimodal datasets with a valid Evidence Lower Bound. Extensive experimental results demonstrate significant improvements of our model over existing methods on various tasks including cross-modal generation, clustering and classification.

## 1 Introduction

Cross-modal representation learning aims to bridge the semantic gap between heterogeneous data from different modalities, such as images, text, audio and video [1, 2, 3]. The key challenge of the task lies in capturing both the modality-specific features and the shared semantic structure across modalities, despite their inherent differences in format and statistical properties [4]. Disentangled representation

---

[*]Both authors contributed equally to this work and are joint first authors.
[†]Corresponding authors.

39th Conference on Neural Information Processing Systems (NeurIPS 2025).

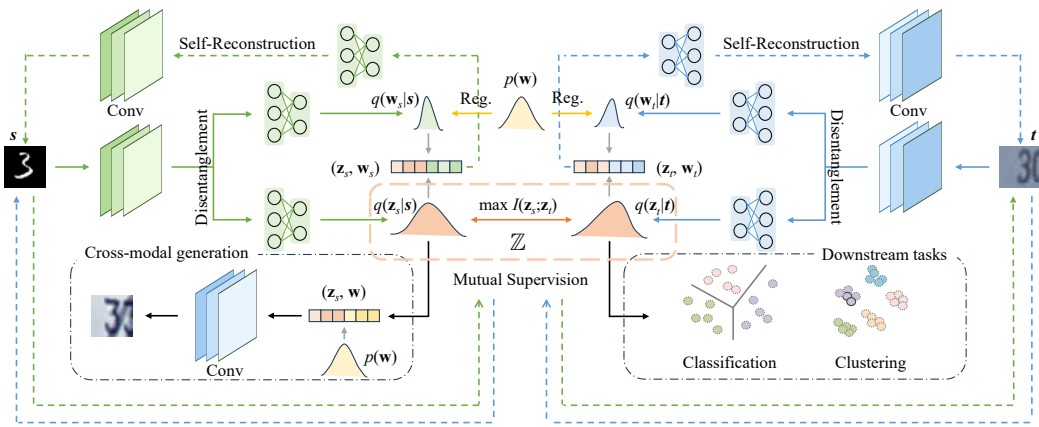

Figure 1: The architecture of DCMEM. For each modality, the encoder extracts shared ($z_s, z_t$) and modality-specific ($w_s, w_t$) latent variables. Self-reconstruction and KL regularization (denoted as "Reg.") ensure disentanglement, where $w$ follows a standard Gaussian prior and shared variables serve as priors for each other. Shared representations lie in a common latent space $\mathbb{Z}$, enabling mutual supervision via cross-modal paths $s \to \mathbb{Z} \to t$ and $t \to \mathbb{Z} \to s$. We further align $z_s$ and $z_t$ by maximizing their mutual information. DCMEM supports cross-modal generation, classification and clustering.

learning has thus emerged as a promising approach to address this challenge by explicitly separating modality-invariant (shared) factors from modality-specific (private) ones, enabling more interpretable and robust cross-modal representations [5]. Variational Autoencoders (VAEs) have been a prevailing framework for disentanglement with their probabilistic formulation and ability to learn structured latent representations. In particular, multimodal VAEs have gained significant attention for their ability to jointly encode and decode information from multiple modalities in a unified probabilistic space [6]. From a methodological perspective, multimodal VAEs can be broadly divided into two categories: models that focus on learning shared latent variables and those that incorporate modality-specific private variables. The former (e.g., MVAE [7], MMVAE [8], MoPoE [9], MEME [10]) aim to capture the common semantic structure across modalities, while the latter (e.g., MMVAE+ [11], CMVAE [12], IIAE [13], Multi-VAE [14]) explicitly disentangle shared and modality-specific information by introducing separate latent spaces. However, these methods either suffer from limited capability to align heterogeneous modalities or lack sufficient structural constraints between modalities, which hinders the clear separation of informative shared and modality-specific factors. In this work, we propose a novel multimodal framework, termed **D**isentangled **C**ross-**M**odal representation learning with **E**nhanced **M**utual supervision (DCMEM), to address these challenges. As illustrated in Figure 1, our model disentangles the common and distinct information across modalities by extracting shared and modality-specific latent representations using multiple VAEs. Specifically, we regularize the shared representation learned from one modality using that from the other in a mutually supervised manner. Moreover, we incorporate the information bottleneck principle into our model to ensure that the shared and modality-specific factors encode exclusive yet complementary information for reconstructing each modality. Finally, we apply an alignment constraint in the shared latent space to promote consistent semantic and inter-modality coherence. Notably, our model is designed to be trainable simultaneously on both complete and partial multimodal datasets with a valid Evidence Lower Bound (ELBO). The main contributions of this paper are summarized as follows:

- We propose DCMEM, a novel multimodal framework that disentangles effectively shared and modality-specific factors. Our model leverages enhanced mutual supervision to improve cross-modal alignment and semantic consistency, while simultaneously enforcing structured representation learning through the information bottleneck principle.

- Our model is inherently designed to be trainable on both complete and partially missing modalities simultaneously. This improves the robustness of our model and broadens its applicability to real-world scenarios where incomplete modalities are prevalent.

- Extensive experiments on three diverse datasets demonstrate that our method produces more coherent and informative embeddings compared to existing multimodal VAE-based approaches

in terms of various evaluation metrics, including generation coherence, clustering accuracy and classification accuracy.

## 2 Related Work

### 2.1 Multimodal VAEs

Multimodal generation tasks have gained significant attention, driving research on various generative models. Among them, multimodal VAEs have become particularly popular due to their impressive performance. Early multimodal VAEs [15, 16, 17] typically employ joint encoders over concatenated inputs, often requiring auxiliary components or processes to support cross-modal generation. Wu *et al.* [7] propose a scalable method based on the Product of Experts (PoE) to address this problem. However, subsequent research [8, 9, 18, 19] identifies several issues with PoE, such as calibration errors. To address these limitations, Shi *et al.* propose MMVAE [8], using Mixture of Experts (MoE) to model the joint posterior as a mixture of unimodal posteriors. The MoPoE model [9] combines PoE and MoE to balance semantic coherence with effective joint distribution learning. Several studies [11, 20] highlight a trade-off between generation quality and coherence in multimodal VAEs. PoE-based models typically achieve lower coherence, while MoE-based models suffer from lower generation quality [11, 21]. To address these issues, various models incorporate different regularizers to improve performance. For instance, MVTCAE [22] uses mutual information theory for regularization, mmJSD [23] employs a dynamic prior to combine modality information, MEME [10] enhances performance through mutual supervision and MMVM [24] regularizes learned posterior approximations with a data-dependent prior. Moreover, Palumbo *et al.* introduce MMVAE+ [11], which incorporates modality-specific subspaces to improve cross-modal likelihood estimation [23, 25]. Building on MMVAE+, CMVAE [12] introduces clustering variables and a mixture distribution for model categories [26, 27], extending multimodal generation to include clustering capabilities. Recently, Gao *et al.* introduce MVP [28] which proposes an informational prior based on cyclic permutations, enabling both generative and clustering tasks.

### 2.2 Information Bottleneck

Information bottleneck plays a crucial role in multimodal clustering and representation learning, enabling models to extract robust and interpretable latent representations by capturing shared and complementary information across different modalities. Wang *et al.* [29] propose a supervised method that maximizes mutual information between joint representations and labels while filtering out irrelevant data from original views. Federici *et al.* introduce MIB [30], which applies the information bottleneck to identify shared and view-specific information between views. Hwang *et al.* [13] develop a cross-domain generative model to enable image-to-image translation, while Lin *et al.* propose COMPLETER [31], which maximizes shared mutual information and minimizes conditional entropy to recover missing views. In addition, CMIB-Nets [32] balances the consistency and complementarity of multimodal views by extracting shared and view-specific information. Wang *et al.* [33] focus on self-supervised learning to regularize mutual information and improve clustering performance. Hu *et al.* [34] introduce a propagation information bottleneck to facilitate the transition from representation learning to clustering structure learning. Huang *et al.* [35] identify key requirements for effective multi-view learning and propose a model that integrates representation learning with clustering. Mao *et al.* [36] enhance alignment in clustering by distinguishing between consistency and redundancy through mutual information maximization, while Yan *et al.* [37] introduce a differentiable information bottleneck for deterministic multi-view clustering.

## 3 Methods

**Problem Statement.** We consider a cross-modal learning scenario where the data consists of two modalities, $s$ and $t$, with some observations potentially missing one of the modalities. In this context, we represent the data containing only modality $s$ as $\mathcal{D}_s$, the data containing only modality $t$ as $\mathcal{D}_t$ and the data containing both modalities paired as $\mathcal{D}_{s,t}$. Our goal is to learn meaningful latent representations from the combined datasets $\mathcal{D} = \mathcal{D}_s \cup \mathcal{D}_t \cup \mathcal{D}_{s,t}$ for various downstream tasks such as clustering and classification, while maintaining high-quality cross-modal generation and coherence.

## 3.1 Disentangled Mutual Supervision

Given the paired data setting $(s, t)$, the information flow $s \leftrightarrow z \leftrightarrow t$ is often employed in cross-modal representation learning, aiming to extract a latent representation $z$ that encodes the information shared between both modalities [10]. However, such a design lacks a separate space for modality-specific information, which causes the shared latent representation $z$ to inevitably contain entangled noise through mutual supervision under both information flows. Therefore, we propose to learn a modality specific factor $w$ along with the shared latent $z$ to decompose the exclusive and common features of each modality. Specifically, we present the main concept by considering the information flow from $s$ to $t$ and the derivation for the opposite direction is similar. According to our assumption, we define the generative process of our model as follows: $p_{\theta, \psi_z}(t, s, z_s, w_s) = p(t) p_{\psi_z}(z_s \mid t) p(w_s) p_\theta(s \mid z_s, w_s)$, where $w_s$ captures modality-specific content from $s$, $z_s$ represents the shared information necessary for generating $t$. The prior of $w_s$ is assumed to follow a standard normal distribution, i.e., $p(w_s) \sim \mathcal{N}(0, \mathbf{I})$. Based on the generative model, the marginal log-likelihood $p_{\theta, \psi_z}(t, s)$ requires integration over the latent variables $z_s$ and $z_w$, which is generally intractable. We then seek a variational posterior $q_{\phi, \varphi}(z_s, w_s | s, t)$ to overcome this intractability and derive the ELBO via Jensen's inequality:

$$\log p_{\theta, \psi_z}(t, s) = \log \int p_{\theta, \psi_z}(t, s, z_s, w_s) \, dz_s dw_s \geq \mathbb{E}_{q_{\phi, \varphi}(z_s, w_s | s, t)} \log \frac{p_{\theta, \psi_z}(t, s, z_s, w_s)}{q_{\phi, \varphi}(z_s, w_s \mid s, t)}. \tag{1}$$

In the course of variational inference, we derive a factorized form of the posterior distribution $q_{\phi, \varphi}(z_s, w_s, t \mid s)$ under the assumptions that (i) the latent variables $z_s$ and $w_s$ are independent, and (ii) the observations $s$ and $t$ are conditionally independent given $z_s$. Thus, the inference process is represented as: $q_{\phi, \varphi}(z_s, w_s, t \mid s) = q_{\phi_z}(z_s \mid s) q_{\phi_w}(w_s \mid s) q_\varphi(t \mid z_s)$. Once $q_{\phi, \varphi}(z_s, w_s, t \mid s)$ is computed, we can apply Bayes' rule to obtain $q_{\phi, \varphi}(z_s, w_s \mid s, t)$ according to the following decomposition: $q_{\phi, \varphi}(z_s, w_s | s, t) = \frac{q_{\phi, \varphi}(z_s, w_s, t | s)}{q_{\phi_z, \varphi}(t | s)} = \frac{q_{\phi_z}(z_s | s) q_{\phi_w}(w_s | s) q_\varphi(t | z_s)}{q_{\phi_z, \varphi}(t | s)}$, where $q_{\phi_z, \varphi}(t \mid s) = \int q_{\phi_z}(z_s \mid s) q_\varphi(t \mid z_s) \, dz_s$. And $\phi = \{\phi_z, \phi_w\}$ denotes the full set of parameters governing the variational distribution of $z_s$ and $w_s$, each modeled as a Gaussian distribution. By substituting inference process into Eq. (1), we obtain:

$$\begin{aligned}
\log p_{\theta, \psi_z}(t, s) \geq \mathbb{E}_{q_\phi(z_s, w_s | s)} & \left[ \frac{q_\varphi(t \mid z_s)}{q_{\phi_z, \varphi}(t \mid s)} \log \frac{p_{\psi_z}(z_s \mid t) p(w_s) p_\theta(s \mid z_s, w_s)}{q_{\phi_z}(z_s \mid s) q_{\phi_w}(w_s \mid s) q_\varphi(t \mid z_s)} \right] \\
& + \log q_{\phi_z, \varphi}(t | s) + \log p(t).
\end{aligned} \tag{2}$$

In Eq. (2), the ratio $\frac{q_\varphi(t | z_s)}{q_{\phi_z, \varphi}(t | s)}$ serves as an importance weight that adjusts for the mismatch between the sampled latent-induced distribution and the actual conditional distribution. This adjustment improves the estimation accuracy and prevents the loss of pivotal information regarding $t$ during sampling. Moreover, the log term measures the agreement between the generative process (through $p_{\psi_z}(z_s \mid t) p(w_s) p_\theta(s \mid z_s, w_s)$) and the variational inference (through $q_{\phi_z}(z_s \mid s) q_{\phi_w}(w_s \mid s) q_\varphi(t \mid z_s)$). Maximizing this term encourages consistency between the latent structure inferred from data and the one induced by the generative process. The additional term $\log q_{\phi_z, \varphi}(t | s)$ acts as a regularization term to encourage its alignment with the true distribution of $t$ and balance the other components of the ELBO. Since $\log p(t)$ is constant with respect to the model parameters, it does not affect the optimization and can be ignored in the objective function.

**Structured Representation Learning.** The maximization of the ELBO alone presented in Eq. (2) is insufficient to ensure complete disentanglement of the latent variables $w_s$ and $z_s$, as any arbitrary mutually exclusive factorization may be equally favored in the absence of mechanisms that explicitly encourage information retention in the shared factor. Considering this, we add a mutual information maximization term $I(z_s; t; s)$ to enforce $z_s$ containing all relevant information across modalties. We further incorporate another mutual information penalty $I(z_s; w_s)$ to encourage the decomposition of $w_s$ and $z_s$. The two information constraints can be unified in the following form:

$$\max I(z_s; t; s) - I(z_s; w_s) = I(s; z_s, w_s) - I(s; w_s) - I(z_s; s \mid t) \tag{3}$$

The detailed derivations for Eq. (3) are provided in the Appendix A.1. Since direct optimization of mutual information is generally intractable, approximation methods such as variational inference or Monte Carlo sampling are often used to estimate these terms [30, 33, 13]. We subsequently derive computationally feasible approximations for each mutual information component as follows.

The first term $I\left(s; z_s, w_s\right)$ quantifies the amount of information about the input modality $s$ that is captured by the latent variables $z_s$ and $w_s$. Since its calculation involves $q(s \mid w_s, z_s) = \frac{q(w_s, z_s \mid s)\, p_D(s)}{\int p_D(s)\, q(z_s, w_s \mid s)\, ds}$, where $p_D(s)$ appears both as the empirical data distribution in the expectation and as an integral term in the denominator, making the computation intractable. We instead obtain a variational lower bound based on the generative distribution $p_\theta\left(s \mid z_s, w_s\right)$:

$$
\begin{aligned}
I\left(z_s, w_s; s\right) &= \mathbb{E}_{q_\phi(z_s, w_s \mid s) p_D(s)} \log \frac{q\left(s \mid z_s, w_s\right)}{p_D\left(s\right)} \\
&= H\left(s\right) + \mathbb{E}_{q_\phi(z_s, w_s \mid s) p_D(s)} \log p_\theta\left(s \mid z_s, w_s\right) \\
&\quad + \mathbb{E}_{q(z_s, w_s)}\left[D_{KL}\left(q\left(s \mid z_s, w_s\right) \| p_\theta\left(s \mid z_s, w_s\right)\right)\right] \\
&\geq H\left(s\right) + \mathbb{E}_{q_\phi(z_s, w_s \mid s) p_D(s)} \log p_\theta\left(s \mid z_s, w_s\right),
\end{aligned}
\tag{4}
$$

where the entropy term $H\left(s\right)$ is treated as a constant. The remaining expectation term corresponds to a reconstruction loss under Gaussian distribution assumption and it indicates that maximizing $I\left(z_s, w_s; s\right)$ facilitates $w_s$ and $z_s$ to jointly contain all relevant information to modality $s$. The second term $-I\left(s; w_s\right)$ can be omitted as a standalone constraint. A detailed explanation is provided in the Appendix A.3. The conditional mutual information term $-I\left(z_s; s \mid t\right)$ is minimized to suppress view-specific redundancy. The latent representation $z_s$ is regularized by the counterpart modality $t$, encouraging $z_s$ to capture only the information accessible from both views. Specifically, by defining $z_s$ and $z_t$ over a shared latent space $\mathbb{Z}$, $I\left(z_s; s \mid t\right)$ can be variationally approximated as:

$$
-I\left(z_s; s \mid t\right) \geq -\mathbb{E}_{p_D(s, t)}\left[D_{KL}\left(q_{\phi_z}\left(z_s = z \mid s\right) \| q_{\psi_z}\left(z_t = z \mid t\right)\right)\right].
\tag{5}
$$

A complete derivation of Eq. (5) is provided in Appendix A.2. Consequently, by integrating these variationally tractable surrogate objectives, namely the reconstruction-based surrogate for $I\left(z_s, w_s; s\right)$ in Eq. (4), the KL divergence bound for $-I\left(s; w_s\right)$ and the cross-modal regularization for $I\left(z_s; s \mid t\right)$ in Eq. (5), the original ELBO in Eq. (2) becomes the following tractable optimization objective:

$$
\begin{aligned}
\mathcal{L}_{\{\boldsymbol{\Phi}, \boldsymbol{\Psi}\}}(s, t) &= \mathbb{E}_{q_\phi(z_s, w_s \mid s)}\left[\frac{q_\varphi\left(t \mid z_s\right)}{q_{\phi_z, \varphi}\left(t \mid s\right)} \log \frac{p_{\psi_z}\left(z_s \mid t\right) p\left(w_s\right) p_\theta\left(s \mid z_s, w_s\right)}{q_{\phi_z}\left(z_s \mid s\right) q_{\phi_w}\left(w_s \mid s\right) q_\varphi\left(t \mid z_s\right)}\right] \\
&\quad + \log q_{\phi_z, \varphi}\left(t \mid s\right) + \mathbb{E}_{q_\phi(z_s, w_s \mid s) p_D(s)} \log p_\theta\left(s \mid z_s, w_s\right) \\
&\quad - \mathbb{E}_{p_D(s, t)}\left[D_{KL}\left(q_{\phi_z}\left(z_s = z \mid s\right) \| q_{\psi_z}\left(z_t = z \mid t\right)\right)\right],
\end{aligned}
\tag{6}
$$

where $\boldsymbol{\Phi} = \{\phi, \varphi\}$ and $\boldsymbol{\Psi} = \{\psi, \theta\}$ correspond to the encoder and decoder parameters, respectively.

**Mutual Supervision.** Mutual supervision leverages reciprocal guidance between modalities to learn a semantically consistent shared latent space. Unlike explicit integration methods like PoE or MoE, it offers greater flexibility and robustness without requiring strict alignment or direct fusion. Building on the notion that each modality both informs and constrains the other, we formulate the objective in the case where $t$ represents the source data and $s$ the target data, i.e. $t \rightarrow z_t \rightarrow s$, as follows:

$$
\begin{aligned}
\mathcal{L}_{\{\boldsymbol{\Psi}, \boldsymbol{\Phi}\}}(s, t) &= \mathbb{E}_{q_\psi(z_t, w_t \mid t)}\left[\frac{q_\theta\left(s \mid z_t\right)}{q_{\theta, \psi_z}\left(s \mid t\right)} \log \frac{p_{\phi_z}\left(z_t \mid s\right) p\left(w_t\right) p_\varphi\left(t \mid z_t, w_t\right)}{q_{\psi_z}\left(z_t \mid t\right) q_{\psi_w}\left(w_t \mid t\right) q_\theta\left(s \mid z_t\right)}\right] \\
&\quad + \log q_{\theta, \psi}\left(s \mid t\right) + E_{q_\psi(z_t, w_t \mid t) p_D(t)} \log p_\varphi\left(t \mid z_t, w_t\right) \\
&\quad - \mathbb{E}_{p_D(s, t)}\left[D_{KL}\left(q_{\psi_z}\left(z_t = z \mid t\right) \| q_{\phi_z}\left(z_s = z \mid s\right)\right)\right].
\end{aligned}
\tag{7}
$$

This formulation mirrors the standard direction $\mathcal{L}_{\{\boldsymbol{\Phi}, \boldsymbol{\Psi}\}}(s, t)$ and captures the reverse information flow. To instantiate this symmetric structure, we implement the model by swapping the roles of the generative and inference networks, where we exchange the parameter sets $\boldsymbol{\Phi}$ and $\boldsymbol{\Psi}$. To integrate both directions of information flow, we combine the contributions from both $\mathcal{L}_{\{\boldsymbol{\Phi}, \boldsymbol{\Psi}\}}(s, t)$ and $\mathcal{L}_{\{\boldsymbol{\Psi}, \boldsymbol{\Phi}\}}(s, t)$, yielding the objective function:

$$
\mathcal{L}_{Bi}\left(s, t\right) = \frac{1}{2}\left(\mathcal{L}_{\{\boldsymbol{\Phi}, \boldsymbol{\Psi}\}}(s, t) + \mathcal{L}_{\{\boldsymbol{\Psi}, \boldsymbol{\Phi}\}}(s, t)\right).
\tag{8}
$$

**Shared Representations Alignment.** We leverage $I(z_s, z_t)$ to directly align the modality-invariant latent variables, preventing them from diverging or encoding discrepant semantics for the same content. This ensures that mutual supervision is not only reflected in the generative reconstruction paths, but also enforced through semantic alignment at the latent representation level. We incorporate

$I(\boldsymbol{z}_s, \boldsymbol{z}_t)$ as a regularization component in addition to Eq. (8) and derive the following training objective in the paired-data scenario:

$$\mathcal{L}_{Bi}(\boldsymbol{s}, \boldsymbol{t}) = \frac{1}{2}\left(\mathcal{L}_{\{\boldsymbol{\Phi}, \boldsymbol{\Psi}\}}(\boldsymbol{s}, \boldsymbol{t}) + \mathcal{L}_{\{\boldsymbol{\Psi}, \boldsymbol{\Phi}\}}(\boldsymbol{s}, \boldsymbol{t})\right) + \alpha I(\boldsymbol{z}_s; \boldsymbol{z}_t). \tag{9}$$

where $\alpha$ serves as a regularization coefficient that balances the contribution of the mutual information term $I(\boldsymbol{z}_s; \boldsymbol{z}_t)$. Empirically, we estimate $I(\boldsymbol{z}_s; \boldsymbol{z}_t)$ using contrastive learning, which has been confirmed as an effective way to solve for the mutual information maximization. Specifically, we align paired latent features while distinguishing unpaired ones, computing pairwise cosine similarities within each batch and applying cross-entropy without extra projection layers.

## 3.2 Partial Observations Scenario

Our model naturally extends to single-view scenarios due to its autoregressive cross-modal generative structure. Given an observed modality, the shared latent representation can be inferred and used to approximate the missing modality via learned generative paths. First considering that the $\boldsymbol{s}$-mode data is available, we can derive a variational approximation for $\log p_{\theta, \psi}(\boldsymbol{s})$ by marginalizing over the unobserved modality $\boldsymbol{t}$, resulting in the following lower bound:

$$\begin{aligned} \log p_{\theta, \psi_z}(\boldsymbol{s}) &= \log \int p(\boldsymbol{t}) \, p_{\psi_z}(\boldsymbol{z}_s \mid \boldsymbol{t}) \, p(\boldsymbol{w}_s) \, p_\theta(\boldsymbol{s} \mid \boldsymbol{z}_s, \boldsymbol{w}_s) \, d\boldsymbol{t} \, d\boldsymbol{z}_s \, d\boldsymbol{w}_s \\ &\geq \mathbb{E}_{q_\phi(\boldsymbol{z}_s, \boldsymbol{w}_s \mid \boldsymbol{s})} \log \frac{p_\theta(\boldsymbol{s} \mid \boldsymbol{z}_s, \boldsymbol{w}_s) \, p(\boldsymbol{w}_s) \, p_{u^t}(\boldsymbol{z}_s)}{q_{\phi_z}(\boldsymbol{z}_s \mid \boldsymbol{s}) \, q_{\phi_w}(\boldsymbol{w}_s \mid \boldsymbol{s})}, \end{aligned} \tag{10}$$

where $p_{u^t}(\boldsymbol{z}_s) = \int p(\boldsymbol{t}) \, p_{\psi_z}(\boldsymbol{z}_s \mid \boldsymbol{t}) \, d\boldsymbol{t}$. Notably, even in the absence of paired $\boldsymbol{t}$-observations, the model can still regularize the latent representation $\boldsymbol{z}_s$ through a shared prior derived from the distribution of $\boldsymbol{t}$. Inspired by VampPrior [38], we define a batch-dependent prior over $B$ representative anchors $\{\boldsymbol{u}_i^t\}_{i=1}^B$ sampled from the $\boldsymbol{t}$-modality, yielding: $p_{u^t}(\boldsymbol{z}_s) = \frac{1}{B}\sum_{i=1}^B p_{\psi_z}(\boldsymbol{z}_s \mid \boldsymbol{u}_i^t)$, where dynamic resampling ensures the prior adapts to the evolving latent structure, improving stability and expressiveness. Eq. (10) can then be rewritten as:

$$\begin{aligned} \mathcal{L}_s(\boldsymbol{s}) &= \mathbb{E}_{q_\phi(\boldsymbol{z}_s, \boldsymbol{w}_s \mid \boldsymbol{s})} \log \frac{p_\theta(\boldsymbol{s} \mid \boldsymbol{z}_s, \boldsymbol{w}_s) \, p(\boldsymbol{w}_s) \frac{1}{B}\sum_{i=1}^B p_{\psi_z}(\boldsymbol{z}_s \mid \boldsymbol{u}_i^t)}{q_{\phi_z}(\boldsymbol{z}_s \mid \boldsymbol{s}) \, q_{\phi_w}(\boldsymbol{w}_s \mid \boldsymbol{s})} \\ &= \mathbb{E}_{q_\phi(\boldsymbol{z}_s, \boldsymbol{w}_s \mid \boldsymbol{s})} \log p_\theta(\boldsymbol{s} \mid \boldsymbol{z}_s, \boldsymbol{w}_s) - D_{KL}\left(q_{\phi_z}(\boldsymbol{z}_s \mid \boldsymbol{s}) \,\|\, \frac{1}{B}\sum_{i=1}^B p_{\psi_z}(\boldsymbol{z}_s \mid \boldsymbol{u}_i^t)\right) \\ &\quad - D_{KL}\left(q_{\phi_w}(\boldsymbol{w}_s \mid \boldsymbol{s}) \,\|\, p(\boldsymbol{w}_s)\right). \end{aligned} \tag{11}$$

With this design, we can still leverage the information from the $\boldsymbol{t}$-modality to effectively constrain and guide the model. In a comparable manner, when the $\boldsymbol{s}$-modality is missing and $\boldsymbol{t}$-modality is available, the objective is defined as:

$$\begin{aligned} \mathcal{L}_t(\boldsymbol{t}) &= \mathbb{E}_{q_\psi(\boldsymbol{z}_t, \boldsymbol{w}_t \mid \boldsymbol{t})} \log p_\varphi(\boldsymbol{t} \mid \boldsymbol{z}_t, \boldsymbol{w}_t) - D_{KL}\left(q_{\psi_z}(\boldsymbol{z}_t \mid \boldsymbol{t}) \,\|\, \frac{1}{B}\sum_{i=1}^B p_{\phi_z}(\boldsymbol{z}_t \mid \boldsymbol{u}_i^s)\right) \\ &\quad - D_{KL}\left(q_{\psi_w}(\boldsymbol{w}_t \mid \boldsymbol{t}) \,\|\, p(\boldsymbol{w}_t)\right). \end{aligned} \tag{12}$$

The final objective function integrates the target loss functions for both paired and unpaired cases, encompassing three scenarios: paired samples, samples with only $\boldsymbol{s}$-modality, and samples with only $\boldsymbol{t}$-modality. It is expressed as follows and the training procedure is detailed in Appendix A.7.

$$\mathcal{L}(\mathcal{D}) = \sum_{s, t \in \mathcal{D}_{s,t}} \mathcal{L}_{Bi}(\boldsymbol{s}, \boldsymbol{t}) + \sum_{s \in \mathcal{D}_s} \mathcal{L}_s(\boldsymbol{s}) + \sum_{t \in \mathcal{D}_t} \mathcal{L}_t(\boldsymbol{t}). \tag{13}$$

# 4 Experiments

## 4.1 Experiments Setup

**Datasets.** Two widely used datasets, i.e. MNIST-SVHN and CUBICC, are adopted in our experiments. The MNIST-SVHN dataset consists of the MNIST and Street View House Numbers (SVHN) datasets, where the samples share digit labels (10 classes) but have different digit styles [8]. The

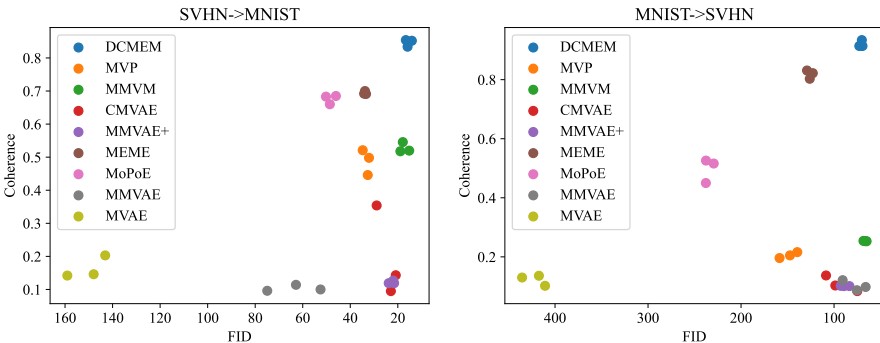

Figure 2: The cross-modal generation performance of DCMEM and existing multimodal VAEs on the MNIST-SVHN dataset. Each model was run independently three times. The best performance is located in the top-right corner of each figure.

CUB Image-Captions for Clustering (CUBICC) dataset, a variant of the CUB image caption dataset constructed by Palumbo *et al.* [12], consists of two modalities: bird images and their corresponding descriptive captions. The dataset is divided into 8 categories based on bird species. In addition, the Human Breast Cancer spatial transcriptomics dataset provides high-resolution spatial measurements across multiple modalities, including gene expression, spatial coordinates, and tissue morphological features [39]. The dataset comprises 20 distinct spatial label categories corresponding to different tissue or structural regions. These datasets are selected as representative benchmarks to evaluate the model's ability to disentangle and align multimodal latent spaces rather than to cover all possible modality combinations.

**Baselines.** To comprehensively evaluate the performance of the proposed method, we compare it with eight existing multimodal VAEs, including MVAE [7], MMVAE [8], MoPoE [9], MEME [10], MMVAE+ [11], CMVAE [12], MMVM [24] and MVP [28]. For all comparison methods, we adopt the model architectures proposed in their respective papers and use their default optimal parameters. To assess the model's capability in handling partially observed datasets, we construct a set of incomplete bimodal datasets by randomly removing one modality at missing rates of $\eta \in \{0.25, 0.5, 0.75\}$ and then train MVAE, MoPoE, MEME, MVP as well as our method on these modified datasets. Each method is run three times to ensure the reliability of the results. We provide the implementation details of our method in Appendix B.2.

**Evaluation.** For the generative task, we primarily evaluate generation coherence and generation quality as in previous work [11, 12, 24]. To evaluate generation coherence, we use a pretrained classifier to classify the generated samples and evaluate the generation coherence in terms of the classification accuracy. The generation quality is assessed using the FID metric [40]. Moreover, we investigate the effectiveness of the learned latent representations based on classification and clustering analyses. For classification, we follow previous work by training a linear classifier on the latent space and report the accuracy. For clustering, we apply $K$-means and evaluate the results using ACC, NMI and ARI. For models with both shared and modality-specific latent variables, all evaluations are conducted on the shared latent space, which captures modality-invariant semantics.

## 4.2 Comparison of Generation Performance

In this section, we compare the performance of the proposed DCMEM model with that of the aforementioned competitive multimodal VAEs in terms of cross-modal generation. We first conduct the cross-modal generation experiment on the fully paired MNIST-SVHN dataset. As shown in Figure 2, our model demonstrates superior performance, achieving both high generative coherence and quality. In Figures 3 and 7, we present the cross-modal generation results for each model. It is evident from these results that our model effectively captures the underlying digit labels and generates accurate cross-modal samples. In contrast, other models tend to misidentify similar digits, struggling with certain digit pairs. This discrepancy highlights the advantages of our model in handling the complexity of cross-modal generation tasks, maintaining both high fidelity and label consistency.

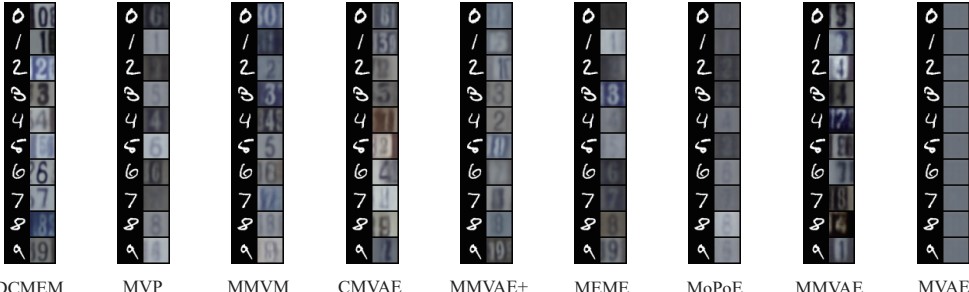

DCMEM    MVP    MMVM    CMVAE    MMVAE+    MEME    MoPoE    MMVAE    MVAE

Figure 3: Qualitative results of cross-modal generation on the MNIST-SVHN dataset for each model. The left side shows the input samples, while the right side displays the cross-modal generated samples.

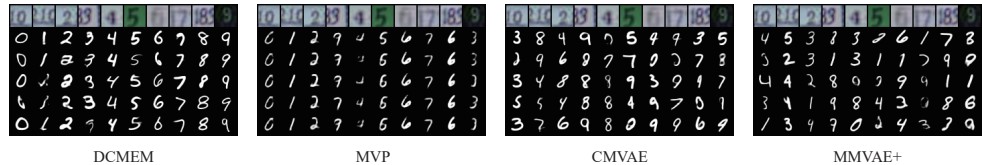

DCMEM        MVP        CMVAE        MMVAE+

Figure 4: Five SVHN-to-MNIST samples are generated by varying only the modality-specific latent variables on the MNIST-SVHN dataset.

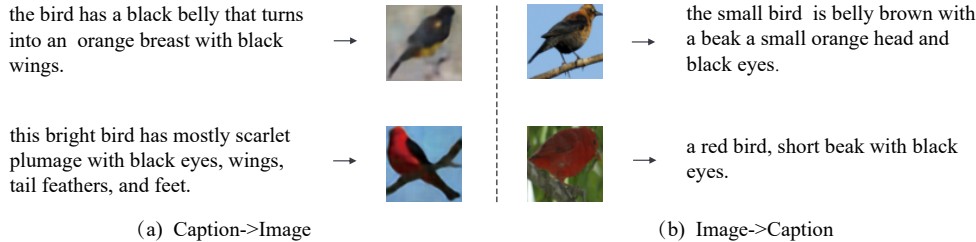

(a) Caption->Image              (b) Image->Caption

Figure 5: Qualitative results of cross-modal generation by DCMEM on the CUBICC dataset. The samples on the left side of the arrows represent the input samples, while the samples on the right side show the generated cross-modal samples.

Among the models, MMVAE+, CMVAE, MVP and our DCMEM all incorporate both shared and modality-specific latent variables. To further analyze the generation coherence and quality, we fix the shared latent variables and randomly sample five different modality-specific latent variables for cross-modal generation. This allows us to evaluate how well each model can generate diverse samples while maintaining consistency across modalities. The experimental results are illustrated in Figures 4 and 8. Our model demonstrates the ability to generate images that maintain the same digit class, while introducing variations in digit shape and color as the modality-specific variables are modified. In contrast, both MMVAE+ and CMVAE exhibit entanglement between class information and modality-specific variables, leading to inconsistencies in the generated samples. Meanwhile, MVP fails to introduce sufficient variation through its modality-specific latent variables, resulting in limited sample diversity. These comparisons further highlight the strength of our approach in producing diverse and high-quality cross-modal generations without sacrificing semantic consistency or class identity.

To further demonstrate the performance of our model in more complex scenarios, we conduct a cross-modal generation experiment on the CUBICC dataset and Figure 5 presents the generation results. As expected, our model consistently achieves high-quality cross-modal generation performance across different modalities. The generated samples align well with the corresponding modalities, maintaining a high level of coherence between the image and text representations. This showcases the model's ability to effectively handle multimodal data in more intricate and diverse settings, demonstrating its robustness in real-world applications. Moreover, the model achieves robust cross-

Table 1: Quantitative comparison of clustering performance for each model's latent representations on the CUBICC dataset. The best and second-best results are highlighted in bold and underlined, respectively.

| Methods | Image Representation | | | Caption Representation | | | Joint Representation | | |
|---|---|---|---|---|---|---|---|---|---|
| | ACC | NMI | ARI | ACC | NMI | ARI | ACC | NMI | ARI |
| MVAE | 26.2 | 12.4 | 7.5 | 18.1 | 2.4 | 0.9 | 38.7 | 26.8 | 18.0 |
| MMVAE | 23.1 | 12.1 | 6.1 | 14.5 | 1.3 | 0.1 | 15.8 | 1.5 | 0.2 |
| MoPoE | 33.4 | 17.6 | 11.5 | 43.5 | 27.1 | 19.9 | 40.8 | 30.4 | 20.2 |
| MEME | 44.8 | 43.4 | 28.4 | 36.3 | 29.5 | 18.6 | 19.8 | 4.8 | 2.1 |
| MMVAE+ | 27.7 | 11.9 | 7.1 | 48.7 | 36.4 | 26.8 | 64.4 | 52.6 | 44.1 |
| CMVAE | 67.7 | 58.3 | 47.4 | 65.1 | **53.3** | 42.7 | 73.7 | 67.4 | 57.2 |
| MMVM | 58.9 | 56.9 | 44.5 | 23.9 | 9.4 | 5.4 | 66.8 | 67.0 | 55.5 |
| MVP | 64.1 | 53.8 | 41.8 | 48.5 | 34.4 | 26.1 | 61.1 | 55.6 | 44.0 |
| DCMEM | **86.9** | **77.4** | **72.4** | **69.7** | 52.2 | **44.2** | **86.3** | **76.8** | **71.5** |

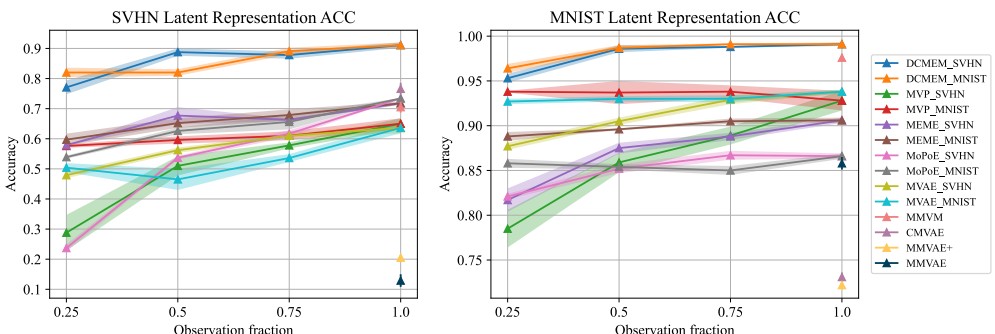

Figure 6: Classification accuracy under different missing rates on the MNIST-SVHN dataset. Shaded areas represent the standard deviation across multiple runs. The subscripts in method names indicate the observed modality. For example, DCMEM_SVHN (Observation fraction = 0.25) denotes that the training data consists of 25% paired samples and 75% unimodal SVHN samples.

modal generation under varying missing rates, effectively balancing generative quality and class-level semantic consistency (see Appendix C.1).

## 4.3 Latent Representation Analysis

In this section, we focus on evaluating the effectiveness of the latent representations learned by the model. All models are trained on the fully paired MNIST-SVHN and CUBICC datasets and the latent representations for the test set are extracted using the trained models. We also evaluate the models under partially paired (incomplete) scenarios, as detailed in Appendix C.2. We first perform clustering on the latent representations of each modality to assess their individual clustering performance, followed by clustering on the joint representations. For models such as MVAE and MoPoE, which produce joint latent representations, we directly use the joint representations for $K$-means clustering. CMVAE directly yields the clustering labels as it learns a clustering variable. For other models that provide neither the joint representations nor clustering factors, we concatenate the latent representations from each modality and then apply $K$-means clustering to the resulting joint representation. The clustering results are summarized in Tables 1 and 2.

Quantitative metrics clearly demonstrate that our model outperforms others in terms of clustering performance. The T-SNE [41] plots of our model's latent representations are presented in Figure 9. From these visualizations, it is evident that the latent representations for both modalities naturally form distinct clusters. Moreover, the latent representations of the same category across modalities align into a single cohesive cluster, indicating that our model effectively captures the shared information between modalities. In contrast, the latent representations of other models either fail to form well-defined clusters or exhibit separation between modalities, as shown in Figures 10 and 11. These results further emphasize the superiority of our model in learning coherent and meaningful latent representations, making it highly effective for clustering tasks involving multimodal data.

For the classification task, we present the results based solely on the classification accuracy metric, evaluating model performance under various missing data conditions. The classification results are shown in Figures 6 and 12. Our model consistently outperforms the other alternatives and demonstrates robust classification performance across varying missing rates. In contrast, other baseline models exhibit significant performance degradation at certain missing rates, indicating the limited capability in exploiting incomplete data. For example, MMVAE+ underperforms on the MNIST-SVHN dataset. This suboptimal performance is likely due to its inability to effectively capture class-discriminative features within the shared latent space. Moreover, although models such as MEME and MVP are designed to handle missing modalities, their performance degrades substantially as the missing rate increases. This suggests a limited robustness to incomplete data, highlighting the advantage of our method in maintaining high classification accuracy under varying degrees of missing information. Analyses in Appendix C.3 and Appendix C.4 further show that our model consistently preserves semantic and class-level alignment across modalities. Appendices C.5–C.7 provide detailed descriptions of the ablation study, parameter analysis, and computational resources.

## 4.4 Application to Human Breast Cancer Dataset

To evaluate the applicability of our model in other fields, we conduct an additional experiment on a spatial transcriptomics dataset of human breast cancer [39]. The dataset presents significant challenges for accurate spatial domain identification due to technical noise and inherent biological variability. We compare our model with eight multimodal VAE baselines and five spatial domain clustering methods, including Scanpy [42], STAGATE [43], GraphST [44], SiGra [45], and xSiGra [46]. For the VAE-based baselines lacking explicit spatial relationships modeling, we adopt the CoordConv strategy [47], encoding spatial (x, y) coordinates as two additional image channels. In contrast, spatial clustering methods are inherently designed to capture spatial dependencies through graph-based or attention mechanisms. As illustrated in Figure 13, DCMEM achieves the highest clustering performance, with an ARI of 55.1 and NMI of 69.7, substantially outperforming both VAE-based and spatially-awared approaches. Among multimodal VAEs, MMVAE achieves the best results, yet it still falls short compared to Scanpy, GraphST and our model. Methods such as MVAE and MMVM struggle to resolve fine-grained cell population structures. Although spatial methods like STAGATE and GraphST generate more coherent partitions, DCMEM produces sharper cluster boundaries and exhibits better alignment with ground truth labels. These results demonstrate the effectiveness of our method in integrating heterogeneous modalities and capturing informative spatial and molecular patterns, underscoring its strong potential for real-world clustering tasks.

# 5 Broader Impact & Limitations

This work focuses on learning disentangled cross-modal representations and enhancing generation quality and coherence by leveraging mutual supervision along with the information bottleneck principle. Our approach is applicable to a variety of tasks in scientific and engineering domains and holds the potential for positive societal impact. In the biomedical field, for example, our model can detect fine-grained cell populations with precise boundaries and identify potential biomarkers associated with tumor heterogeneity for subsequent clinical validation, by analyzing spatial transcriptomics data at the single-cell level. At the bulk cancer omics level, it can also assist in identifying novel disease subtypes and predicting the survival outcomes to inform better treatment strategy design. Although our model yields better representations and more coherent outputs, it is specifically designed for bimodal data scenarios and may require non-trivial effort to achieve competitve performance on general multimodal datasets.

# 6 Conclusion

In this work, we propose DCMEM, a variational framework that integrates disentanglement and mutual supervision to learn structured cross-modal representations. It separates shared and modality-specific information via dedicated latent spaces and promotes semantic alignment by maximizing mutual information between shared latent variables across modalities, achieving strong performance across diverse tasks and settings.

## Acknowledgements

This work was supported by the National Natural Science Foundation of China (62372279, 62322215, 62532017), and the Natural Science Foundation of Shandong Province (ZR2025QB62, ZR2023MF119). This study was also supported in part by the High-Performance Computing Center of Central South University.

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

# A  Supplementary Technical Details

## A.1  Mutual Information Decomposition for Structured Representation Learning

Define $X, Y$ and $Z$ be random variables. The the chain rule for mutual information is:

$$I(X;Y;Z) = I(X;Y) - I(X;Y|Z) = I(X;Z) - I(X;Z|Y) = I(Y;Z) - I(Y;Z|X) \quad (14)$$

As defined in the main text, the modality-invariant variable $\boldsymbol{z}_s$ and the modality-specific variable $\boldsymbol{w}_s$ have already been introduced. $I(\boldsymbol{z}_s; \boldsymbol{w}_s)$ and $I(\boldsymbol{s}; \boldsymbol{t}; \boldsymbol{z}_s)$ can be computed as follows:

$$\begin{aligned}
I(\boldsymbol{z}_s; \boldsymbol{w}_s) &= I(\boldsymbol{z}_s; \boldsymbol{s}) - I(\boldsymbol{z}_s; \boldsymbol{s} \mid \boldsymbol{w}_s) + I(\boldsymbol{w}_s; \boldsymbol{z}_s \mid \boldsymbol{s}) \\
I(\boldsymbol{s}; \boldsymbol{t}; \boldsymbol{z}_s) &= I(\boldsymbol{z}_s; \boldsymbol{s}) - I(\boldsymbol{z}_s; \boldsymbol{s} \mid \boldsymbol{t})
\end{aligned} \quad (15)$$

Under the assumption that the modality-invariant variable $\boldsymbol{z}_s$ and the modality-specific variable $\boldsymbol{w}_s$ are conditionally independent given the input $\boldsymbol{s}$, we have: $q(\boldsymbol{w}_s \mid \boldsymbol{s}) = q(\boldsymbol{w}_s \mid \boldsymbol{s}, \boldsymbol{z}_s)$. Thus, the conditional mutual information $I(\boldsymbol{w}_s; \boldsymbol{z}_s \mid \boldsymbol{s})$ simplifies to $I(\boldsymbol{w}_s; \boldsymbol{z}_s \mid \boldsymbol{s}) = H(\boldsymbol{w}_s \mid \boldsymbol{s}) - H(\boldsymbol{w}_s \mid \boldsymbol{s}, \boldsymbol{z}_s) = 0$. As a result, $I(\boldsymbol{w}_s; \boldsymbol{z}_s)$ can be further decomposed as follows: $I(\boldsymbol{z}_s; \boldsymbol{w}_s) = I(\boldsymbol{z}_s; \boldsymbol{s}) - I(\boldsymbol{z}_s; \boldsymbol{s} \mid \boldsymbol{w}_s) = I(\boldsymbol{z}_s; \boldsymbol{s}) + I(\boldsymbol{s}; \boldsymbol{w}_s) - I(\boldsymbol{s}; \boldsymbol{z}_s, \boldsymbol{w}_s)$. In conclusion, the mutual information decomposition for disentangled representation learning is given as follows:

$$\begin{aligned}
&I(\boldsymbol{z}_s; \boldsymbol{t}; \boldsymbol{s}) - I(\boldsymbol{z}_s; \boldsymbol{w}_s) \\
&= \cancel{I(\boldsymbol{z}_s; \boldsymbol{s})} - I(\boldsymbol{z}_s; \boldsymbol{s} \mid \boldsymbol{t}) + I(\boldsymbol{s}; \boldsymbol{z}_s, \boldsymbol{w}_s) - I(\boldsymbol{s}; \boldsymbol{w}_s) - \cancel{I(\boldsymbol{z}_s; \boldsymbol{s})} \\
&= I(\boldsymbol{s}; \boldsymbol{z}_s, \boldsymbol{w}_s) - I(\boldsymbol{s}; \boldsymbol{w}_s) - I(\boldsymbol{z}_s; \boldsymbol{s} \mid \boldsymbol{t})
\end{aligned} \quad (16)$$

## A.2  Evidence Lower Bound on the cross-modal regularization

Assuming that both $\boldsymbol{z}_t$ and $\boldsymbol{z}_s$ lie in the same latent space $\mathbb{Z}$:

$$\begin{aligned}
-I(\boldsymbol{z}_s; \boldsymbol{s} \mid \boldsymbol{t}) &= -\mathbb{E}_{p_D(\boldsymbol{s},\boldsymbol{t})} \mathbb{E}_{q_{\phi_z}(\boldsymbol{z}_s \mid \boldsymbol{s})} \left[ \log \frac{q_{\phi_z}(\boldsymbol{z}_s = \boldsymbol{z} \mid \boldsymbol{s})}{p_{\psi_z}(\boldsymbol{z}_s = \boldsymbol{z} \mid \boldsymbol{t})} \right] \\
&= -\mathbb{E}_{p_D(\boldsymbol{s},\boldsymbol{t})} \mathbb{E}_{q_{\phi_z}(\boldsymbol{z}_s \mid \boldsymbol{s})} \left[ \log \frac{q_{\phi_z}(\boldsymbol{z}_s = \boldsymbol{z} \mid \boldsymbol{s})}{q_{\psi_z}(\boldsymbol{z}_t = \boldsymbol{z} \mid \boldsymbol{t})} \frac{q_{\psi_z}(\boldsymbol{z}_t = \boldsymbol{z} \mid \boldsymbol{t})}{p_{\psi_z}(\boldsymbol{z}_s = \boldsymbol{z} \mid \boldsymbol{t})} \right] \\
&= -\mathbb{E}_{p_D(\boldsymbol{s},\boldsymbol{t})} \left[ D_{KL}(q_{\phi_z}(\boldsymbol{z}_s = \boldsymbol{z} \mid \boldsymbol{s}) \| q_{\psi_z}(\boldsymbol{z}_t = \boldsymbol{z} \mid \boldsymbol{t})) \right] \\
&\quad + \mathbb{E}_{p_D(\boldsymbol{t})} \left[ D_{KL}(p_{\psi_z}(\boldsymbol{z}_s = \boldsymbol{z} \mid \boldsymbol{t}) \| q_{\psi_z}(\boldsymbol{z}_t = \boldsymbol{z} \mid \boldsymbol{t})) \right] \\
&\geq -\mathbb{E}_{p_D(\boldsymbol{s},\boldsymbol{t})} \left[ D_{KL}(q_{\phi_z}(\boldsymbol{z}_s = \boldsymbol{z} \mid \boldsymbol{s}) \| q_{\psi_z}(\boldsymbol{z}_t = \boldsymbol{z} \mid \boldsymbol{t})) \right].
\end{aligned} \quad (17)$$

Similarly, $-I(\boldsymbol{z}_t; \boldsymbol{t} \mid \boldsymbol{s})$ can be computed as:

$$-I(\boldsymbol{z}_t; \boldsymbol{t} \mid \boldsymbol{s}) \geq -\mathbb{E}_{p_D(\boldsymbol{s},\boldsymbol{t})} \left[ D_{KL}(q_{\psi_z}(\boldsymbol{z}_t = \boldsymbol{z} \mid \boldsymbol{t}) \| q_{\phi_z}(\boldsymbol{z}_s = \boldsymbol{z} \mid \boldsymbol{s})) \right]. \quad (18)$$

## A.3  Derivation of the Mutual Information Term $I(\boldsymbol{s}; \boldsymbol{w}_s)$

The second term $-I(\boldsymbol{s}; \boldsymbol{w}_s)$ also poses computational challenges due to its dependence on the marginal distribution $p_D(\boldsymbol{s})$, where $q(\boldsymbol{w}_s) = \int q(\boldsymbol{w}_s \mid \boldsymbol{s}) p_D(\boldsymbol{s}) d\boldsymbol{s}$. In a similar vein, this term is approximated using its variational lower bound $-\mathbb{E}_{p_D(\boldsymbol{s})} \left[ D_{KL}(q_{\phi_w}(\boldsymbol{w}_s \mid \boldsymbol{s}) \| p(\boldsymbol{w}_s)) \right]$. This KL divergence term naturally appears in the ELBO objective (Eq. (2)) due to $\mathbb{E}_{q_\phi(\boldsymbol{z}_s, \boldsymbol{w}_s \mid \boldsymbol{s})} \left[ \frac{q_\varphi(\boldsymbol{t} \mid \boldsymbol{z}_s)}{q_{\phi_z, \varphi}(\boldsymbol{t} \mid \boldsymbol{s})} \log \frac{p(\boldsymbol{w}_s)}{q_{\phi_w}(\boldsymbol{w}_s \mid \boldsymbol{s})} \right] = \mathbb{E}_{q_{\phi_w}}(\boldsymbol{w}_s \mid \boldsymbol{s}) \log \frac{p(\boldsymbol{w}_s)}{q_{\phi_w}(\boldsymbol{w}_s \mid \boldsymbol{s})} = -\mathbb{E}_{p_D(\boldsymbol{s})} \left[ D_{KL}(q_{\phi_w}(\boldsymbol{w}_s \mid \boldsymbol{s}) \| p(\boldsymbol{w}_s)) \right]$. Hence, it is implicitly optimized through the ELBO and can be omitted as a standalone constraint.

## A.4  Mutual Information Approximation

The mutual information $I(\boldsymbol{z}_s, \boldsymbol{z}_t)$ between the shared latent variables $\boldsymbol{z}_s$ and $\boldsymbol{z}_t$ is approximated using a contrastive loss that encourages alignment between representations of paired inputs while distinguishing those from non-paired ones [48]. Given a batch of $B$ paired latent features $(\boldsymbol{h}_s, \boldsymbol{h}_t)$ sampled from $\boldsymbol{z}_s$ and $\boldsymbol{z}_t$, we concatenate them into a set of $2B$ vectors. Pairwise cosine similarities are computed and scaled by a fixed temperature $\tau = 0.5$ to construct the contrastive logits. Positive pairs are defined between the $i$-th feature in $\boldsymbol{h}_s$ and the $i$-th feature in $\boldsymbol{h}_t$, as they correspond to the paired input instances. All other pairs in the batch are treated as negatives. We apply a cross-entropy loss to encourage the model to assign higher similarity to positives than to negatives.

### A.5 Cross-Modal Reconstruction Mechanism

In our framework, $q_\varphi(t \mid z_s)$ serves a key role in the mutual supervision mechanism. Specifically, $z_s$ is the shared latent representation inferred from modality $s$ and $q_\varphi(t \mid z_s)$ models the reconstruction of modality $t$ based solely on this shared information. This setup is intentionally designed to exclude the private latent variable $w$, since the goal is to assess what information is common and transferable across modalities. In practice, we implement $q_\varphi(t \mid z_s)$ by setting $w = 0$ and passing the concatenation $[z_s, 0]$ into the decoder of modality $t$. This design offers two key advantages: (1) it avoids introducing an additional decoder by reusing $p_\varphi(t \mid z_s, w)$ with $w$ set to zero, keeping the architecture compact; and (2) it promotes effective disentanglement, as reconstructing $t$ from $z_s$ alone forces $z_s$ to capture modality-invariant, shared information.

### A.6 Importance Sampling Stability

The importance sampling weight $\frac{q_\varphi(t|z_s)}{q_{\phi_z,\varphi}(t|s)}$ in Equation 2 plays a critical role in our training objective, but its direct computation can introduce numerical instability due to high variance in gradient estimates. This arises because both the numerator and denominator are parameterized distributions that are learned during training, and their stochastic nature may result in noisy, unreliable updates when used in Monte Carlo estimation of the ELBO. To address this issue, we adopt a stop-gradient strategy to stabilize training without compromising the objective, following a rationale similar to that in prior work [49, 10]. Concretely, we prevent gradients from flowing through the $\frac{q_\varphi(t|z_s)}{q_{\phi_z,\varphi}(t|s)}$, treating it as a fixed scalar during backpropagation: stop_gradient $\left(\frac{q_\varphi(t|z_s)}{q_{\phi_z,\varphi}(t|s)}\right)$. This modification ensures that the parameters are optimized based on more stable signals, as it avoids amplifying gradient noise through the ratio. Importantly, this treatment does not alter the forward computation of the objective but improves the robustness and reliability of the training dynamics. In summary, this design provides a practical and effective solution to variance-induced instability in training, while still aligning with the theoretical intent of the original ELBO formulation.

### A.7 Training procedure for DCMEM

---

**Algorithm 1:** Optimization Procedure of DCMEM

---

**Input:** Multimodal dataset: $\mathcal{D} = \mathcal{D}_s \cup \mathcal{D}_t \cup \mathcal{D}_{s,t}$; Training epochs number: $M$;
       Hyperparameters: $\alpha$; Model parameters $\{\Phi, \Psi\}$
**Output:** Latent representation $z_s$ and $z_t$

1   Randomly initialize model parameters $\{\Phi, \Psi\}$;
2   **for** *epoch $< M$* **do**
3      **for** *each sample in paired data $\mathcal{D}_{s,t}$* **do**
4         Compute $\mathcal{L}_{\{\Phi,\Psi\}}(s, t)$ and $\mathcal{L}_{\{\Psi,\Phi\}}(s, t)$ via Eqs. (6) and (7);
5         Estimate enhanced mutual-supervised information regularization by Eq. (8);
6         Compute the bidirectional lower bound $\mathcal{L}_{\text{Bi}}(s, t)$ by Eq. (9);
7      **for** *each sample in modality-specific data $\mathcal{D}_s$* **do**
8         Compute $\mathcal{L}_s(s)$ via Eq. (11);
9      **for** *each sample in modality-specific data $\mathcal{D}_t$* **do**
10        Compute $\mathcal{L}_t(t)$ via Eq. (12);
11      Update parameters $\{\Phi, \Psi\}$ by maximizing the overall objective in Eq. (13);
12   Compute latent representations $z_s = f_{\phi_z}(s)$ and $z_t = f_{\psi_z}(t)$ using the optimized parameters;
13   **return** *Latent representationas $z_s$ and $z_t$*

---

## B Dataset and Implementation Details

### B.1 Dataset Licences

- MNIST_SVHN: originally published in [8], downloaded the data from http://yann.lecun.com/exdb/mnist, http://ufldl.stanford.edu/housenumbers and the code from https://github.com/iffsid/mmvae, licensed under GPL 3.0.

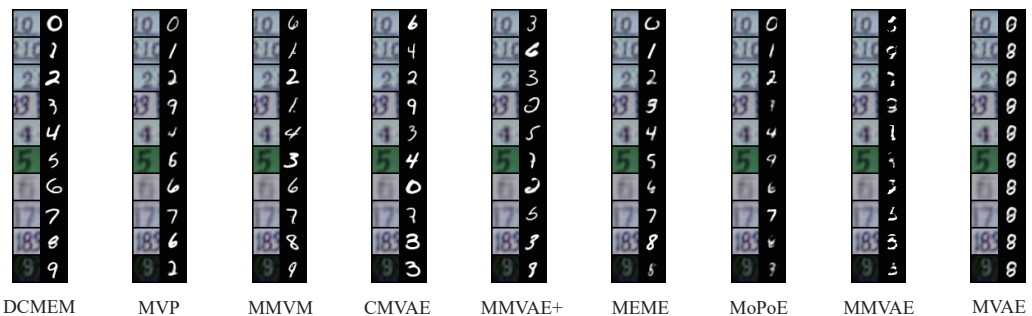

| DCMEM | MVP | MMVM | CMVAE | MMVAE+ | MEME | MoPoE | MMVAE | MVAE |

Figure 7: Supplementary qualitative results of cross-modal generation on the MNIST-SVHN dataset for each model.

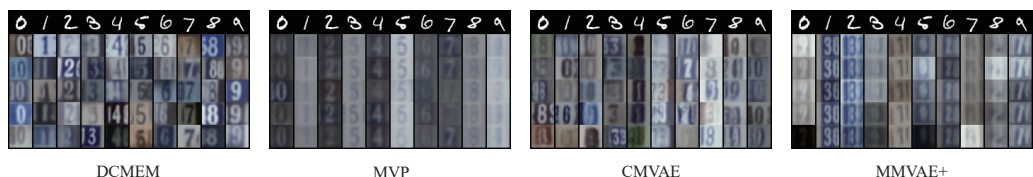

| DCMEM | MVP | CMVAE | MMVAE+ |

Figure 8: Five MNIST-to-SVHN samples are generated by varying only the modality-specific latent variables on the MNIST-SVHN dataset.

- CUBICC: originally published in [12], downloaded the data from https://polybox.ethz.ch/index.php/s/LRkTC2oa6YHHlUj/download, published under the MIT license.
- Human Breast Cancer: originally published in [39], downloaded the data from https://www.10xgenomics.com/datasets/human-breast-cancer-block-a-section-1-1-standard-1-0-0, published under the CC BY 4.0 license.

## B.2 Implementation Details

To ensure a fair comparison, all models are run on a local server with an NVIDIA GeForce RTX 2080 Ti GPU, 64 GB of RAM running Ubuntu 18.04. For our model, we use a ResNet encoder and decoder for image data, and convolutional encoders and decoders for text data. The parameter $\alpha$ is set to 1. For the MNIST-SVHN dataset, the dimensions of the shared and specific latent spaces are set to 32. We use the Adam optimizer with a learning rate of 5e-4, a batch size of 64 and train the model for 100 epochs. For the CUBICC dataset, the dimensions of the shared and specific latent spaces are set to 48 and 16, respectively. The Adam optimizer is used with a learning rate of 1e-4, a batch size of 16 and training is conducted for 200 epochs. For the spatial transcriptomics dataset, we preprocess the gene expression data by selecting the top 3000 highly variable genes, followed by standard normalization and log-transformation. Both the encoder and decoder for this modality are implemented as fully connected neural networks. For the tissue morphology modality, input images are resized to 128×128 pixels. To incorporate spatial context, we adopt the CoordConv [47] technique by appending the 2D spatial coordinates (x, y) as two additional input channels, resulting in a 5-channel input. This modality is processed using convolutional neural networks for both encoding and decoding. Both the shared and specific latent dimensions are set to 32. Optimization is performed using Adam with a learning rate of 5e-4, a batch size of 64 and 100 training epochs.

## C Additional Experimental Results

### C.1 Additional Results on Cross-Modal Generation

To further evaluate the generative capability of our model, we conduct additional experiments under varying missing rates, focusing on both generative coherence and quality. The quantitative results

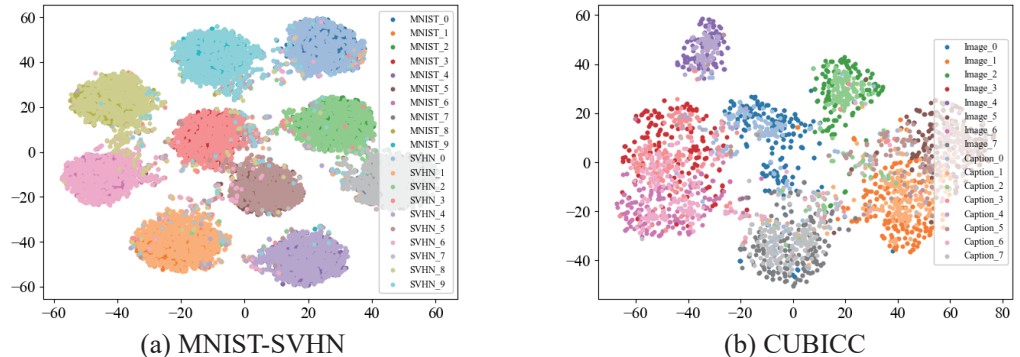

(a) MNIST-SVHN                    (b) CUBICC

Figure 9: T-SNE plot of the latent representations obtained by DCMEM on the MNIST-SVHN and CUBICC datasets. Here, MNIST_0 represents the data from the MNIST modality with the digit label 0, and similarly for other labels.

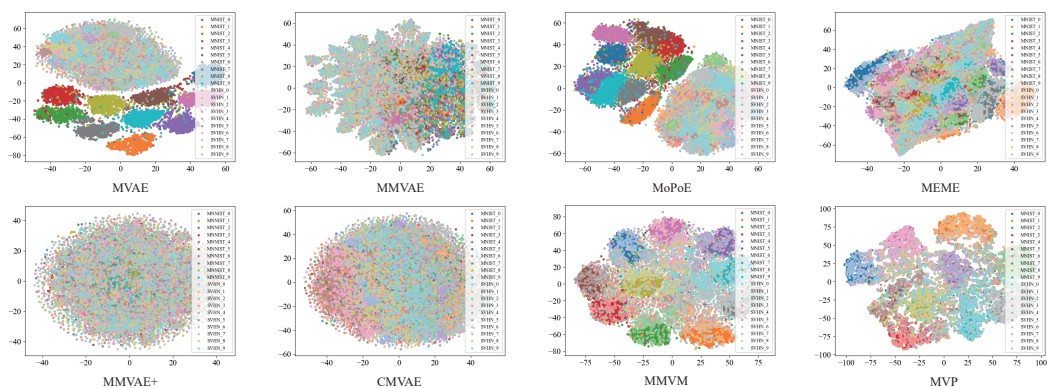

Figure 10: T-SNE plot of the latent representations obtained by baseline models on the MNIST-SVHN dataset.

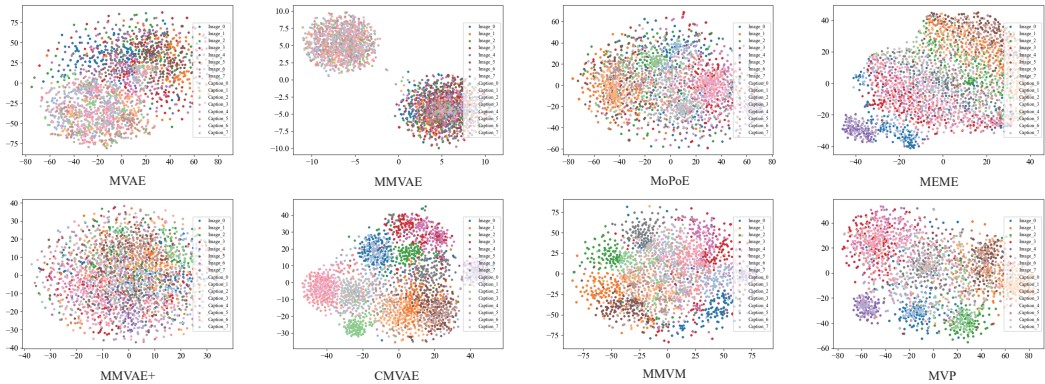

Figure 11: T-SNE plot of the latent representations obtained by baseline models on the CUBICC dataset.

on the MNIST-SVHN dataset are shown in Figures 14 and 15, while the qualitative results are presented in Figures 16–22. As shown in the figures, our model consistently achieves robust cross-modal generation under different levels of missing data. Although a slight degradation in generative consistency is observed as the missing rate increases, the model still maintains strong performance and demonstrates competitive stability compared to other baselines.

Table 2: Quantitative comparison of clustering performance for each model's latent representations on the MNIST-SVHN dataset. The best and second-best results are highlighted in bold and underlined, respectively.

| Methods | SVHN Representation | | | MNIST Representation | | | Joint Representation | | |
|---|---|---|---|---|---|---|---|---|---|
| | ACC | NMI | ARI | ACC | NMI | ARI | ACC | NMI | ARI |
| MVAE | 27.9 | 16.0 | 13.1 | 79.2 | 65.5 | 62.6 | 42.7 | 35.3 | 24.5 |
| MMVAE | 22.0 | 10.4 | 10.1 | 21.8 | 10.3 | 10.1 | 22.6 | 10.7 | 10.1 |
| MoPoE | 37.9 | 27.2 | 18.5 | 50.5 | 45.6 | 33.0 | 64.1 | 60.5 | 50.7 |
| MEME | 21.9 | 10.3 | 10.0 | 36.5 | 32.1 | 20.4 | 22.4 | 10.6 | 10.1 |
| MMVAE+ | 23.9 | 11.4 | 11.1 | 21.3 | 10.4 | 10.0 | 22.9 | 11.9 | 10.8 |
| CMVAE | 42.2 | 36.3 | 25.4 | 28.1 | 15.9 | 14.5 | 32.3 | 19.5 | 15.4 |
| MMVM | 42.2 | 27.1 | 20.7 | 88.1 | 82.1 | 80.4 | 77.5 | 72.2 | 67.5 |
| MVP | 53.6 | 38.7 | 30.1 | 81.4 | 79.6 | 73.6 | 84.8 | 76.4 | 70.6 |
| DCMEM | **91.5** | **80.6** | **82.0** | **99.1** | **97.3** | **98.0** | **99.5** | **98.4** | **98.9** |

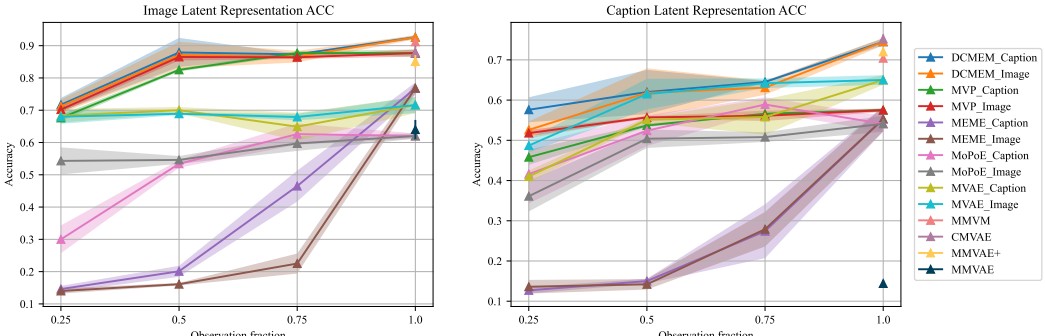

Figure 12: Classification accuracy under different missing rates on the CUBICC dataset. Shaded areas represent the standard deviation across multiple runs. The subscripts in method names indicate the observed modality. For example, DCMEM_Caption (Observation fraction = 0.25) denotes that the training data consists of 25% paired samples and 75% unimodal Caption samples.

Similarly, the cross-modal generation results on the CUBICC dataset are shown in Figures 23 and 24. Our model achieves the highest generative coherence across all missing conditions, indicating its effectiveness in preserving class-level semantics despite incomplete input. However, we observe a moderate decline in generation quality. This trade-off is primarily due to the inherent tension between the generation and clustering objectives: while generation benefits from latent representations that retain fine-grained modality-specific details, clustering prefers representations that focus on global class-level features. As a result, our model strategically balances these two objectives rather than optimizing solely for one, which inevitably limits performance in either direction when pursued independently.

## C.2 Additional Results on Clustering

We evaluate the clustering performance of MVAE, MoPoE, MEME, MVP and DCMEM under varying missing scenarios on both the MNIST-SVHN and CUBICC datasets. As shown in Tables 3, 4, 5 and 6, we consider two settings for each dataset: one where the first modality is partially missing (e.g., MNIST or Image), and one where the second modality is partially missing (e.g., SVHN or Caption). For each setting, clustering is performed based on the latent representations learned from individual modalities as well as their joint embedding. DCMEM consistently achieves the best performance across all settings and representation types, demonstrating strong robustness to incomplete data. Even at low paired data fractions (e.g., 25%), DCMEM maintains high clustering accuracy, while baseline models such as MVAE, MoPoE, MEME and MVP exhibit significant performance degradation. This is especially evident in the Caption modality of the CUBICC dataset, where several baselines struggle to learn meaningful representations under high missing-view rates. Overall, these results highlight the effectiveness of DCMEM in learning coherent and discriminative

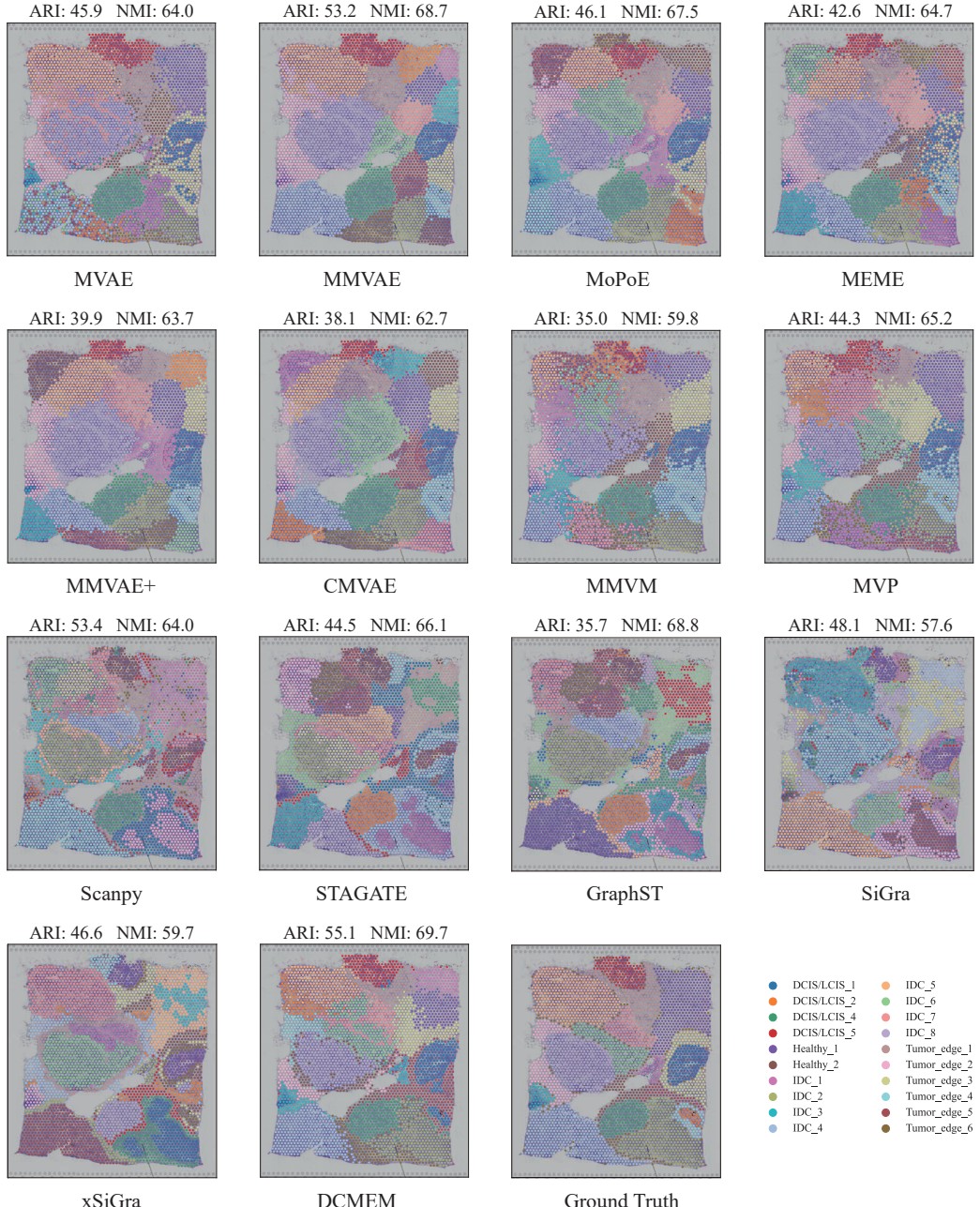

Figure 13: Visualization of clustering results on the human breast cancer dataset. Each subplot shows the clustering output of a different method, with colors indicating predicted clusters. Each method is run three times, and the mean ARI and NMI scores are reported above each plot.

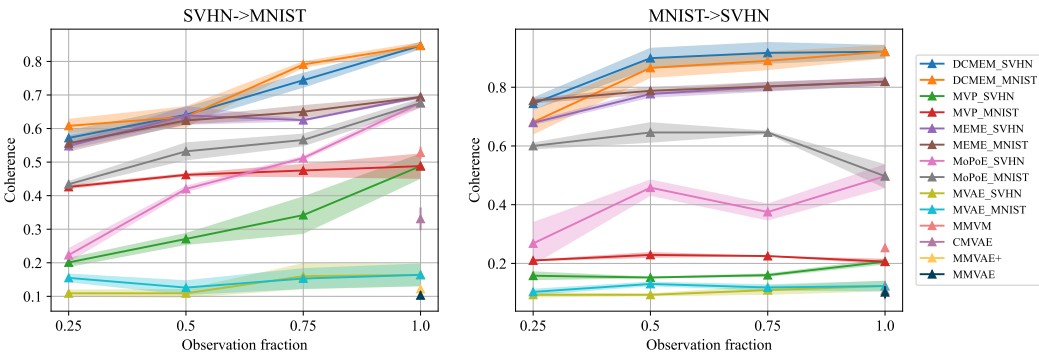

Figure 14: Classification accuracy of cross-modal generations under different missing rates on the MNIST-SVHN dataset. Shaded areas represent the standard deviation across multiple runs. The subscripts in method names indicate the observed modality. For example, DCMEM_SVHN (Observation fraction = 0.25) denotes that the training data consists of 25% paired samples and 75% unimodal SVHN samples.

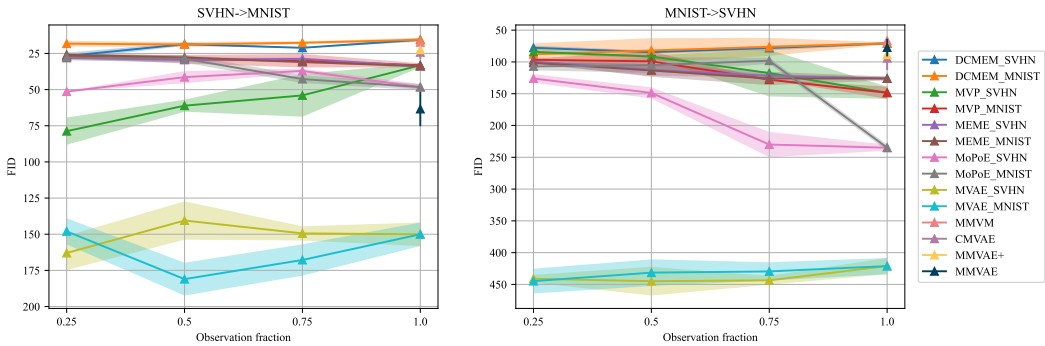

Figure 15: FID score of cross-modal generations under different missing rates on the MNIST-SVHN dataset. Shaded areas represent the standard deviation across multiple runs. The subscripts in method names indicate the observed modality. For example, DCMEM_SVHN (Observation fraction = 0.25) denotes that the training data consists of 25% paired samples and 75% unimodal SVHN samples.

latent spaces across different datasets and under various levels of modality incompleteness. Its ability to leverage both paired and unimodal data allows it to maintain superior clustering performance, setting it apart from existing multimodal VAE approaches.

## C.3 Semantic Relatedness in the Latent Space

Semantic relatedness refers to the notion that semantically aligned multimodal inputs should yield more similar latent distributions than unrelated pairs. To investigate whether our models as well as the baselines exhibit this behavior, we adopt the 2-Wasserstein distance as a measure of semantic similarity between latent distributions. This metric is well-suited for comparing Gaussian distributions due to its closed-form expression in such cases. In our experiment, we compute pairwise 2-Wasserstein distances between all combinations of latent distributions within a mini-batch. We then visualize the resulting distances using histograms, color-coded to distinguish paired samples from unpaired samples. A clear separation between the two groups in the histogram indicates that the model captures meaningful semantic alignment across modalities. Figure 25 illustrates the relatedness histograms produced by our model on the MNIST-SVHN and CUBICC datasets. Figures 26 and 27 show the results for the baseline models.

As shown in Figure 25, our model exhibits consistently lower 2-Wasserstein distances for paired samples, while unpaired samples yield significantly higher distances. This clear separation demonstrates that our model effectively captures semantic alignment across modalities. In contrast, baseline models

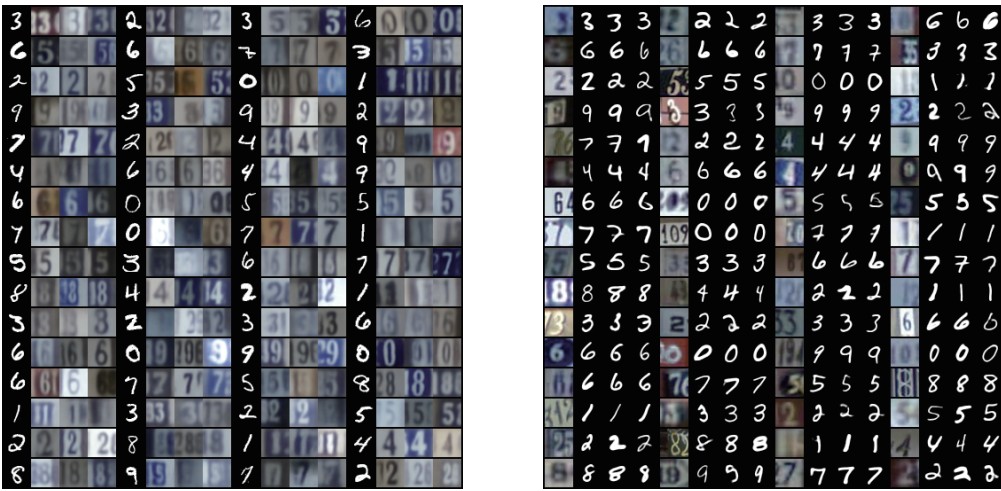

Figure 16: MNIST->SVHN (Left) and SVHN->MNIST (Right), for the fully observed case.

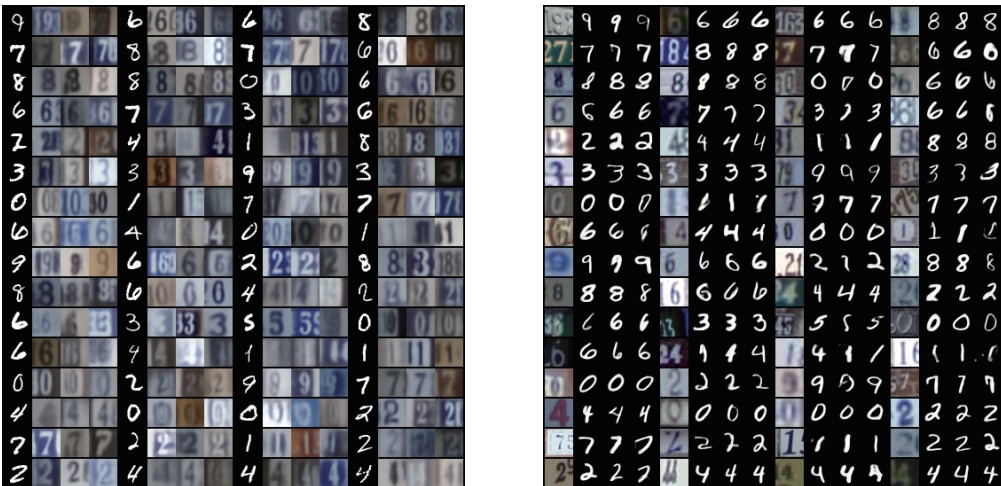

Figure 17: MNIST->SVHN (Left) and SVHN->MNIST (Right), when MNIST is observed 75% of the time.

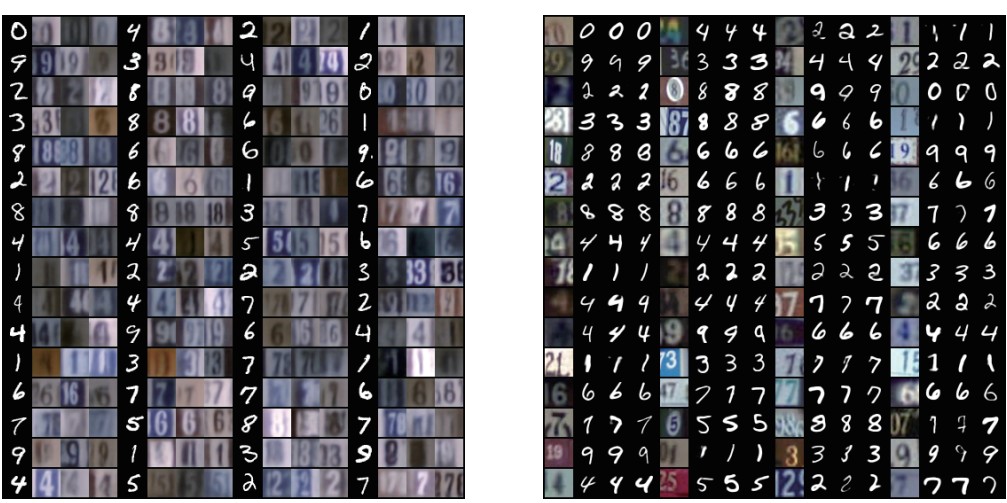

Figure 18: MNIST->SVHN (Left) and SVHN->MNIST (Right), when SVHN is observed 75% of the time.

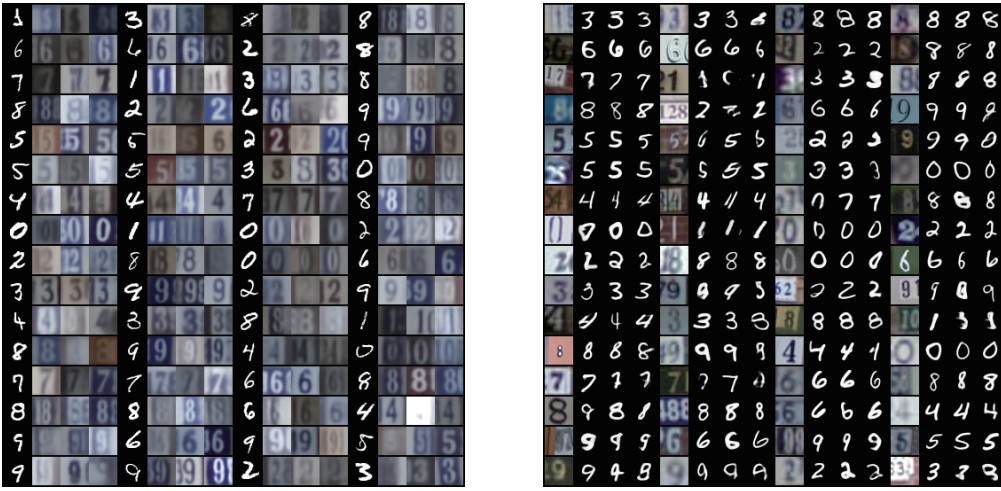

Figure 19: MNIST->SVHN (Left) and SVHN->MNIST (Right), when MNIST is observed 50% of the time.

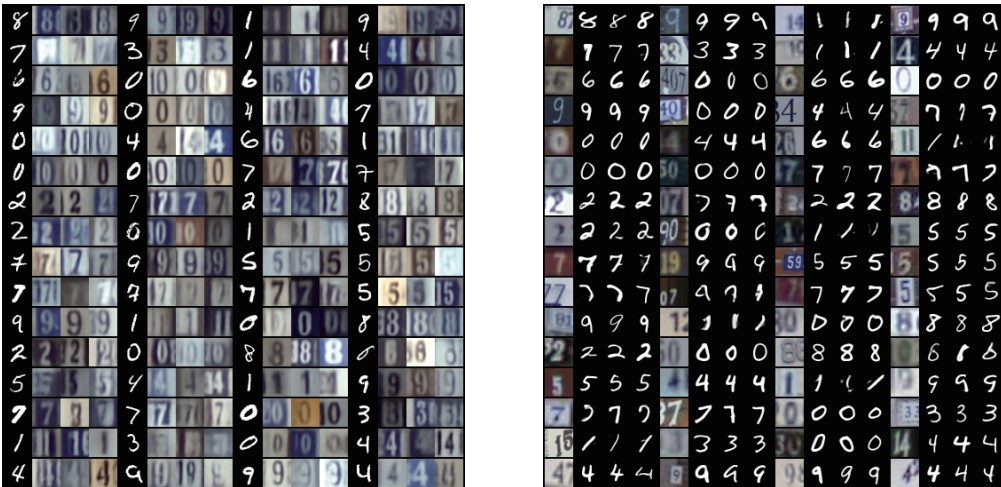

Figure 20: MNIST->SVHN (Left) and SVHN->MNIST (Right), when SVHN is observed 50% of the time.

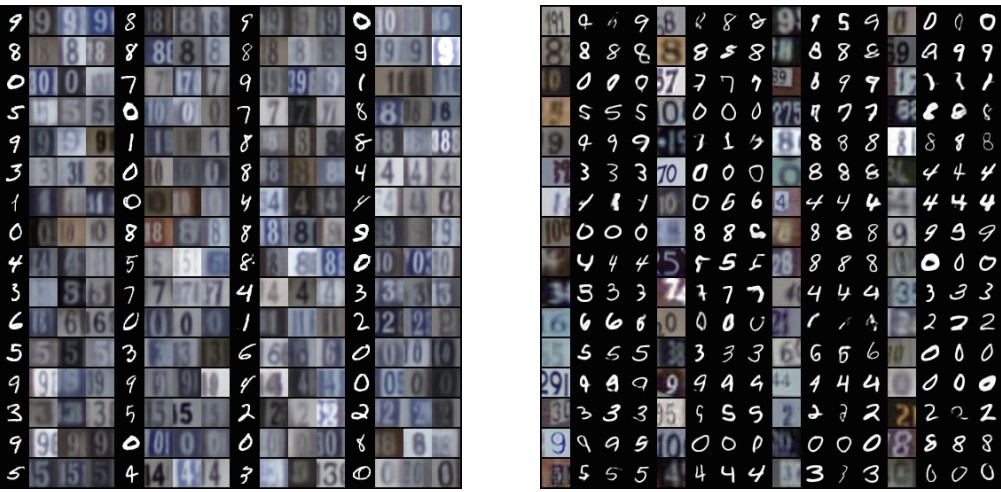

Figure 21: MNIST->SVHN (Left) and SVHN->MNIST (Right), when MNIST is observed 25% of the time.

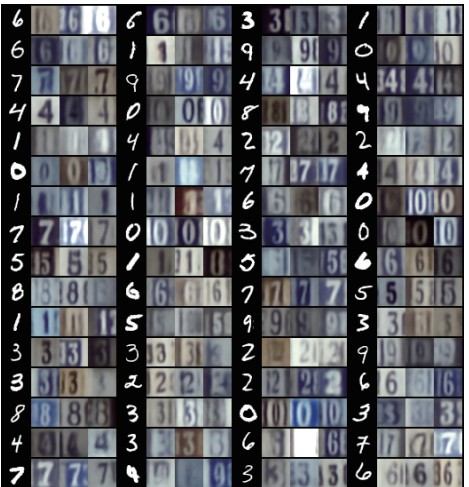 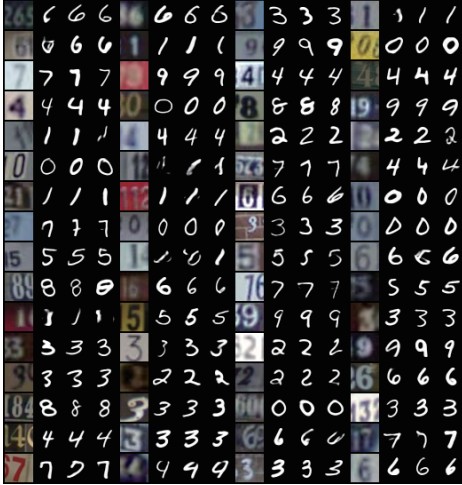

Figure 22: MNIST->SVHN (Left) and SVHN->MNIST (Right), when SVHN is observed 25% of the time.

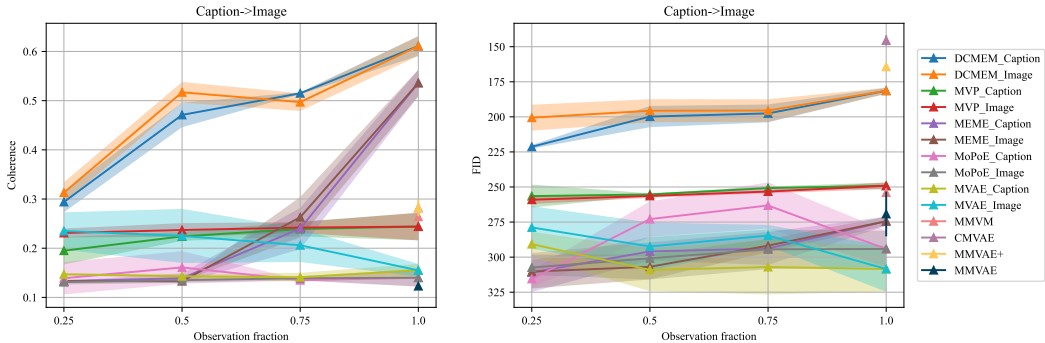

Figure 23: Classification accuracy and FID score of cross-modal generations under different missing rates on the CUBICC dataset. Shaded areas represent the standard deviation across multiple runs. The subscripts in method names indicate the observed modality. For example, DCMEM_Caption (Observation fraction = 0.25) denotes that the training data consists of 25% paired samples and 75% unimodal Caption samples.

such as MVAE and MMVAE display similar distance distributions for both paired and unpaired data, suggesting that their latent representations fail to encode meaningful semantic information. Although MoPoE and MEME capture a certain degree of semantic relatedness, as evidenced by the relatively small gap between paired and unpaired distributions, they achieve weaker semantic alignment compared to our model. MMVAE+, CMVAE and MMVM are only able to capture semantic differences on a single dataset with limited generalization capability. Although MVP shows a noticeable separation, the contrast between paired and unpaired distances is less pronounced than in our model. These comparisons further highlight the superior ability of our approach to learn semantically structured and modality-aligned latent representations.

## C.4  Class-Contextual Relatedness in the Latent Space

To evaluate whether our model captures class-level semantic alignment across modalities, we conduct a class-contextual relatedness analysis on both the MNIST-SVHN and CUBICC datasets. Following the methodology proposed in MEME [10], we compute a class-conditioned distance matrix $K \in \mathbb{R}^{C \times C}$, where $C$ is the number of classes in the dataset. Each entry $K_{ij}$ represents the average 2-Wasserstein distance between the latent distributions of class $i$ from one modality and class $j$ from the other. Ideally, if the model successfully aligns class-level semantics across modalities, we expect the matrix to exhibit low distances along the diagonal (representing matched classes) and higher

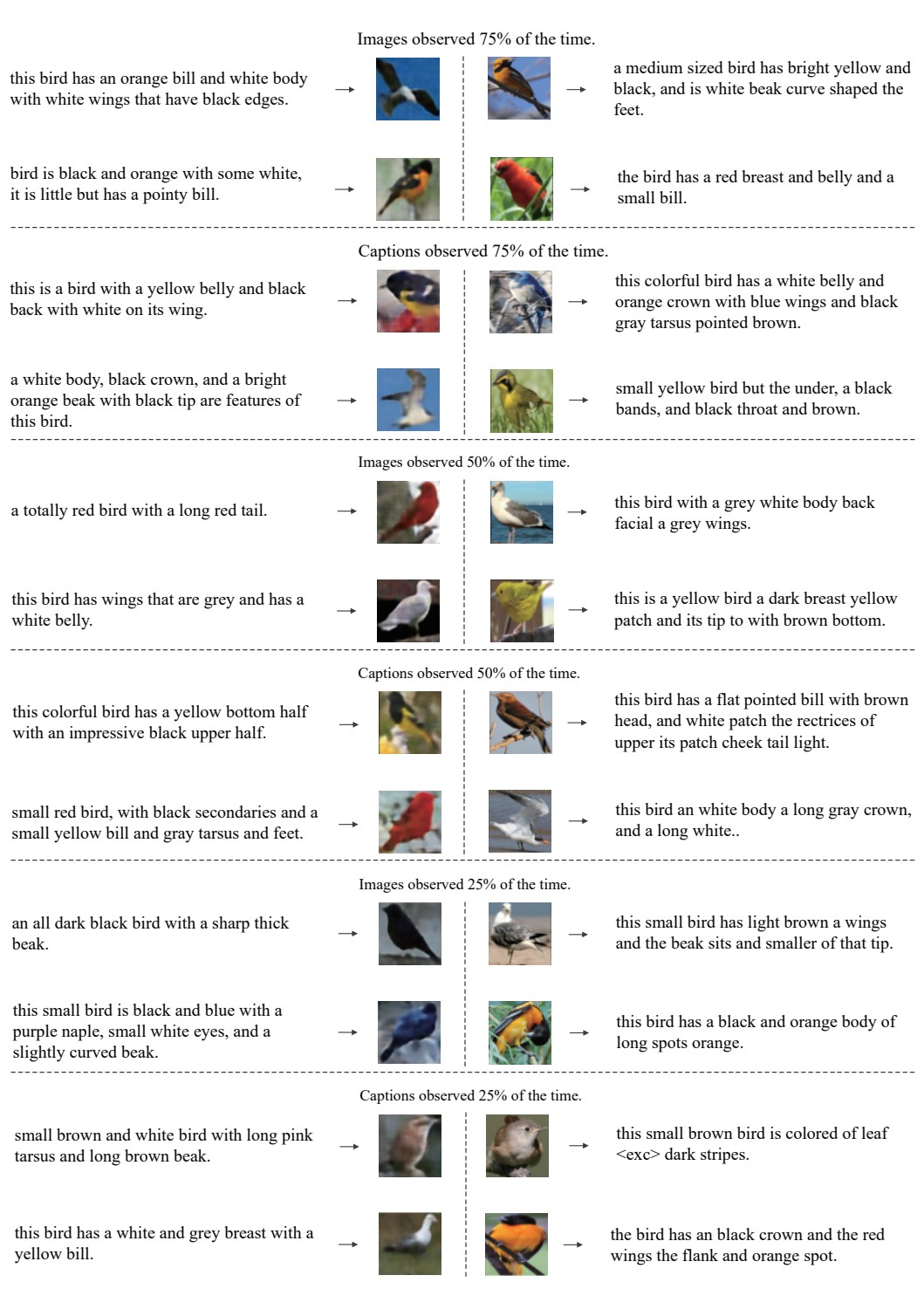

Images observed 75% of the time.

this bird has an orange bill and white body with white wings that have black edges. → → a medium sized bird has bright yellow and black, and is white beak curve shaped the feet.

bird is black and orange with some white, it is little but has a pointy bill. → → the bird has a red breast and belly and a small bill.

Captions observed 75% of the time.

this is a bird with a yellow belly and black back with white on its wing. → → this colorful bird has a white belly and orange crown with blue wings and black gray tarsus pointed brown.

a white body, black crown, and a bright orange beak with black tip are features of this bird. → → small yellow bird but the under, a black bands, and black throat and brown.

Images observed 50% of the time.

a totally red bird with a long red tail. → → this bird with a grey white body back facial a grey wings.

this bird has wings that are grey and has a white belly. → → this is a yellow bird a dark breast yellow patch and its tip to with brown bottom.

Captions observed 50% of the time.

this colorful bird has a yellow bottom half with an impressive black upper half. → → this bird has a flat pointed bill with brown head, and white patch the rectrices of upper its patch cheek tail light.

small red bird, with black secondaries and a small yellow bill and gray tarsus and feet. → → this bird an white body a long gray crown, and a long white..

Images observed 25% of the time.

an all dark black bird with a sharp thick beak. → → this small bird has light brown a wings and the beak sits and smaller of that tip.

this small bird is black and blue with a purple naple, small white eyes, and a slightly curved beak. → → this bird has a black and orange body of long spots orange.

Captions observed 25% of the time.

small brown and white bird with long pink tarsus and long brown beak. → → this small brown bird is colored of leaf <exc> dark stripes.

this bird has a white and grey breast with a yellow bill. → → the bird has an black crown and the red wings the flank and orange spot.

Figure 24: Cross-modal generations on the CUBICC dataset by DCMEM.

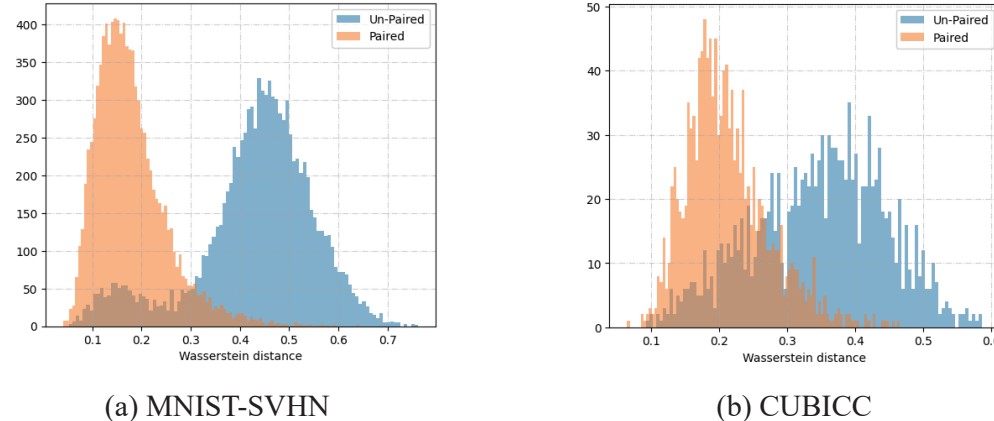

(a) MNIST-SVHN                      (b) CUBICC

Figure 25: Histograms of 2-Wasserstein distances between latent distributions for paired and unpaired multimodal samples obtained by DCMEM on the MNIST-SVHN and CUBICC datasets.

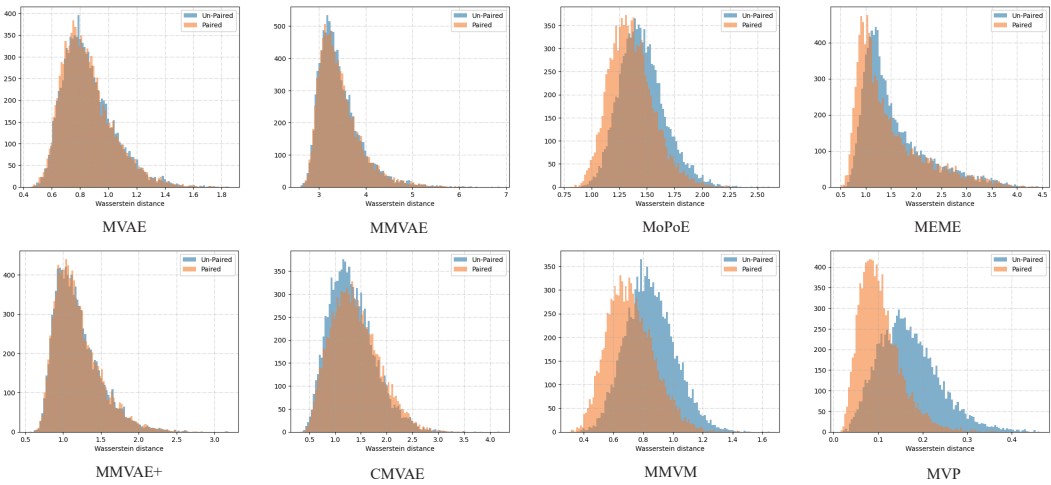

Figure 26: Histograms of 2-Wasserstein distances between latent distributions for paired and unpaired multimodal samples obtained by baseline models on the MNIST-SVHN dataset.

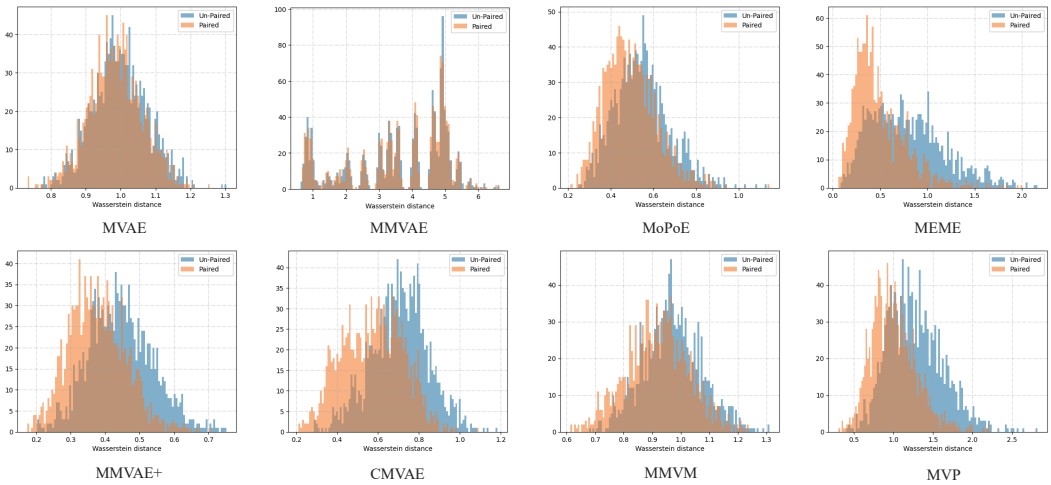

Figure 27: Histograms of 2-Wasserstein distances between latent distributions for paired and unpaired multimodal samples obtained by baseline models on the CUBICC dataset.

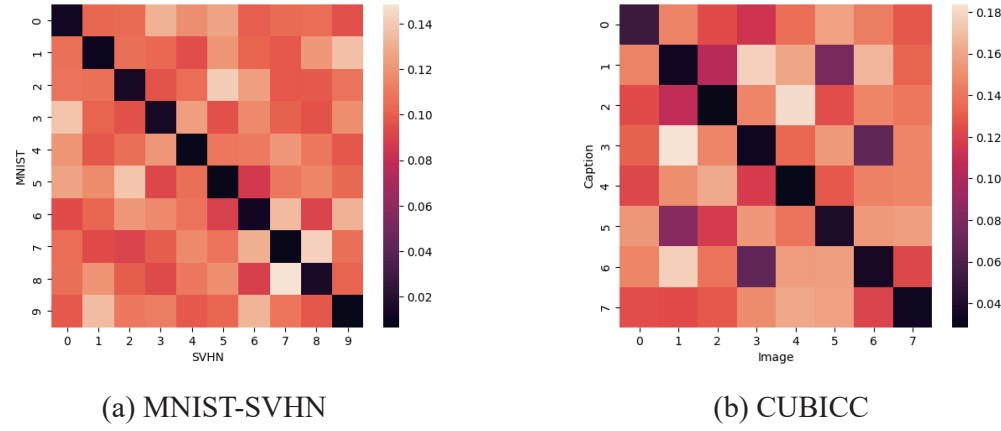

(a) MNIST-SVHN           (b) CUBICC

Figure 28: Heatmaps of class-conditioned 2-Wasserstein distances between latent distributions obtained by DCMEM on the MNIST-SVHN and CUBICC datasets.

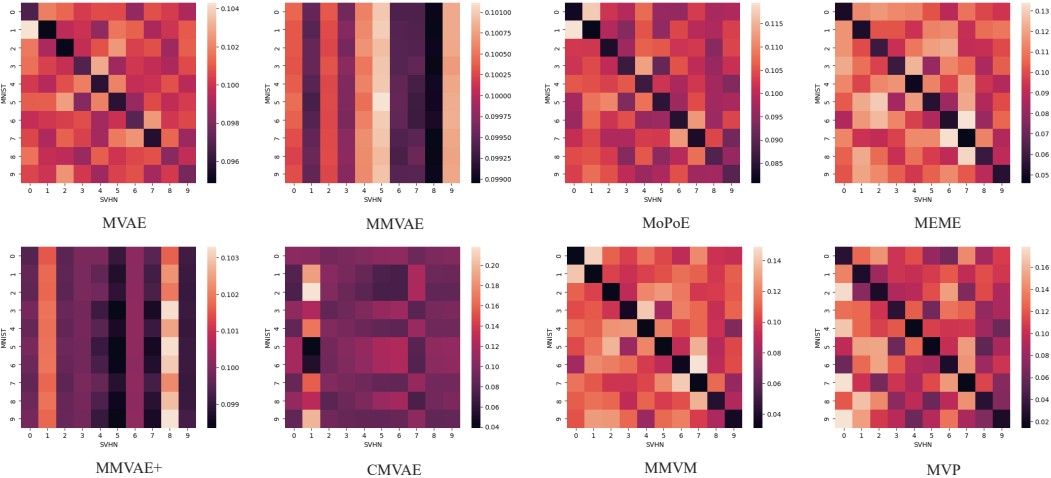

Figure 29: Heatmaps of class-conditioned 2-Wasserstein distances between latent distributions obtained by baseline models on the MNIST-SVHN dataset.

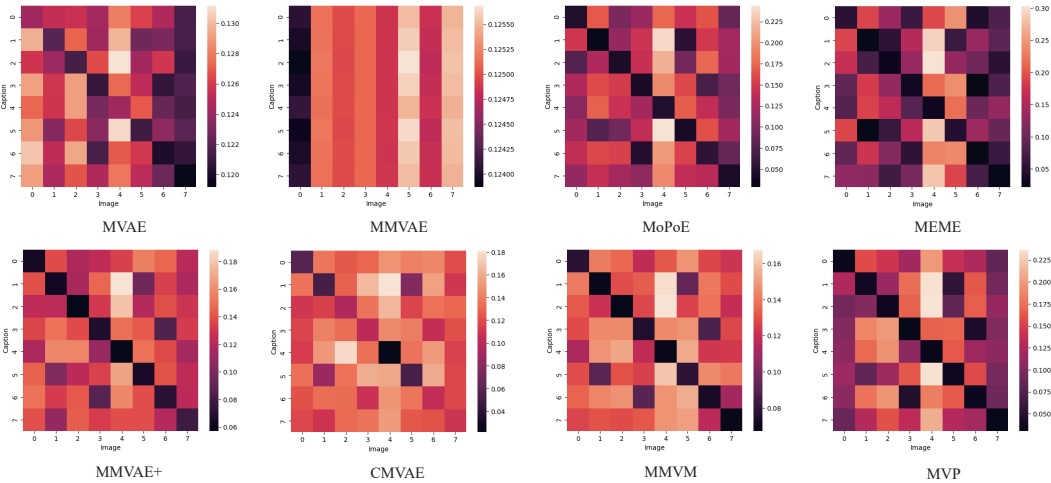

Figure 30: Heatmaps of class-conditioned 2-Wasserstein distances between latent distributions obtained by baseline models on the CUBICC dataset.

Table 3: Quantitative comparison of clustering performance based on latent representations under different missing rates in the MNIST modality on the MNIST-SVHN dataset. Fraction indicates the proportion of paired samples relative to the full training set. Each model is trained on a dataset consisting of paired data at a proportion of Fraction and unimodal SVHN data at a proportion of 1-Fraction, and evaluated on the complete test set.

| Fraction | Methods | SVHN Representation | | | MNIST Representation | | | Joint Representation | | |
|---|---|---|---|---|---|---|---|---|---|---|
| | | ACC | NMI | ARI | ACC | NMI | ARI | ACC | NMI | ARI |
| | MVAE | 18.0 | 6.0 | 3.1 | 77.5 | 63.4 | 59.8 | 43.3 | 41.9 | 28.1 |
| | MoPoE | 27.0 | 13.4 | 8.3 | 50.3 | 41.4 | 29.0 | 61.8 | 52.4 | 42.7 |
| 0.75 | MEME | 11.8 | 0.3 | 0.1 | 41.6 | 35.9 | 24.2 | 11.7 | 0.4 | 0.1 |
| | MVP | 56.4 | 54.6 | 39.9 | 61.3 | 60.3 | 45.7 | 37.8 | 37.6 | 22.1 |
| | DCMEM | **87.4** | **73.8** | **74.0** | **98.7** | **96.3** | **97.2** | **99.5** | **98.4** | **98.9** |
| | MVAE | 18.0 | 6.0 | 3.2 | 70.6 | 56.0 | 52.1 | 43.3 | 41.8 | 27.7 |
| | MoPoE | 22.5 | 10.0 | 5.8 | 52.4 | 45.8 | 34.9 | 52.8 | 45.4 | 34.3 |
| 0.5 | MEME | 11.7 | 0.3 | 0.1 | 41.8 | 35.8 | 24.6 | 11.6 | 0.3 | 0.1 |
| | MVP | 29.0 | 16.0 | 10.2 | 50.6 | 48.8 | 33.2 | 42.9 | 39.0 | 26.7 |
| | DCMEM | **87.5** | **76.1** | **75.8** | **98.3** | **95.2** | **96.2** | **98.4** | **96.6** | **97.5** |
| | MVAE | 17.9 | 5.9 | 3.1 | 68.3 | 56.5 | 50.4 | 39.3 | 33.7 | 22.4 |
| | MoPoE | 18.6 | 6.1 | 3.3 | 54.3 | 43.5 | 31.8 | 42.5 | 35.9 | 23.9 |
| 0.25 | MEME | 11.6 | 0.3 | 0.1 | 45.8 | 36.9 | 27.0 | 11.7 | 0.3 | 0.1 |
| | MVP | 12.3 | 0.6 | 0.1 | 39.1 | 42.0 | 24.8 | 12.3 | 0.6 | 0.1 |
| | DCMEM | **62.0** | **47.2** | **38.4** | **92.3** | **83.4** | **90.3** | **92.3** | **83.4** | **84.0** |

Table 4: Quantitative comparison of clustering performance based on latent representations under different missing rates in the SVHN modality on the MNIST-SVHN dataset. Fraction indicates the proportion of paired samples relative to the full training set. Each model is trained on a dataset consisting of paired data at a proportion of Fraction and unimodal MNIST data at a proportion of 1-Fraction, and evaluated on the complete test set.

| Fraction | Methods | SVHN Representation | | | MNIST Representation | | | Joint Representation | | |
|---|---|---|---|---|---|---|---|---|---|---|
| | | ACC | NMI | ARI | ACC | NMI | ARI | ACC | NMI | ARI |
| | MVAE | 17.2 | 4.8 | 2.6 | 82.4 | 68.0 | 65.7 | 40.3 | 34.6 | 23.9 |
| | MoPoE | 32.1 | 21.6 | 13.4 | 53.8 | 47.2 | 35.5 | 61.1 | 52.4 | 42.9 |
| 0.75 | MEME | 13.0 | 3.8 | 2.2 | 39.3 | 36.8 | 24.1 | 13.4 | 6.6 | 3.3 |
| | MVP | 49.0 | 31.0 | 23.9 | 78.1 | 71.0 | 64.9 | 75.9 | 70.9 | 64.0 |
| | DCMEM | **89.0** | **76.2** | **77.2** | **99.1** | **97.3** | **98.0** | **99.7** | **99.1** | **99.4** |
| | MVAE | 16.4 | 5.5 | 2.4 | 79.6 | 65.4 | 62.5 | 38.6 | 38.4 | 24.7 |
| | MoPoE | 31.1 | 18.5 | 11.6 | 54.8 | 47.2 | 36.3 | 59.4 | 49.6 | 41.9 |
| 0.5 | MEME | 14.5 | 3.3 | 1.6 | 39.6 | 33.5 | 22.1 | 12.3 | 0.6 | 0.2 |
| | MVP | 49.7 | 33.1 | 24.5 | 83.3 | 72.9 | 69.9 | 77.4 | 73.5 | 67.3 |
| | DCMEM | **85.5** | **72.4** | **72.7** | **98.7** | **96.3** | **97.1** | **99.6** | **98.8** | **99.1** |
| | MVAE | 17.1 | 5.4 | 2.7 | 76.9 | 63.7 | 60.6 | 38.3 | 31.1 | 20.7 |
| | MoPoE | 25.7 | 13.0 | 7.4 | 65.3 | 53.0 | 46.3 | 54.6 | 44.0 | 34.3 |
| 0.25 | MEME | 15.8 | 4.9 | 2.6 | 51.1 | 43.0 | 34.4 | 20.5 | 13.0 | 7.7 |
| | MVP | 47.6 | 49.5 | 33.5 | 85.7 | 71.3 | 68.4 | 87.2 | 76.9 | 76.6 |
| | DCMEM | **82.6** | **65.6** | **65.4** | **95.2** | **88.4** | **89.6** | **98.5** | **96.0** | **96.8** |

values off-diagonal (mismatched classes). To visualize this, we present the resulting matrices as heatmaps in Figure 28, where darker colors indicate smaller distances. The baseline results are shown in Figures 29 and 30.

As shown in Figure 28, our model produces a clear diagonal structure in the class-conditioned distance matrices, indicating that it effectively aligns semantically corresponding classes across modalities. This pattern is consistently observed on both the MNIST-SVHN and CUBICC datasets, suggesting robust class-level semantic alignment in the learned latent space. In comparison, baseline models such as MVAE, MMVAE+ and CMVAE fail to exhibit a clear diagonal on at least one of the datasets, revealing their limited ability to consistently model class-level correspondence. Other models, including MEME and MVP, either produce diagonals with less pronounced contrast or show

Table 5: Quantitative comparison of clustering performance based on latent representations under different missing rates in the Image modality on the CUBICC dataset. Fraction indicates the proportion of paired samples relative to the full training set. Each model is trained on a dataset consisting of paired data at a proportion of Fraction and unimodal Caption data at a proportion of 1-Fraction, and evaluated on the complete test set.

| Fraction | Methods | Image Representation | | | Caption Representation | | | Joint Representation | | |
|---|---|---|---|---|---|---|---|---|---|---|
| | | ACC | NMI | ARI | ACC | NMI | ARI | ACC | NMI | ARI |
| | MVAE | 27.2 | 11.8 | 7.0 | 17.8 | 2.1 | 0.7 | 36.9 | 27.7 | 18.7 |
| | MoPoE | 38.1 | 28.4 | 17.4 | 50.8 | 37.0 | 27.5 | 58.5 | 46.6 | 35.2 |
| 0.75 | MEME | 45.8 | 42.1 | 25.0 | 27.6 | 16.3 | 8.5 | 45.0 | 37.1 | 24.1 |
| | MVP | 53.1 | 46.6 | 34.8 | 44.3 | 30.2 | 21.6 | 72.7 | 61.7 | 51.1 |
| | DCMEM | **83.6** | **72.4** | **66.3** | **62.6** | **43.9** | **35.0** | **84.1** | **73.5** | **67.0** |
| | MVAE | 27.0 | 14.3 | 7.7 | 17.7 | 2.4 | 0.9 | 36.4 | 22.1 | 13.6 |
| | MoPoE | 32.6 | 20.2 | 12.4 | 46.2 | 26.6 | 17.0 | 42.1 | 32.6 | 23.4 |
| 0.5 | MEME | 20.4 | 4.8 | 2.4 | 16.9 | 1.4 | 0.2 | 19.8 | 4.2 | 1.8 |
| | MVP | 58.7 | 45.4 | 36.4 | 31.0 | 18.5 | 13.0 | 52.4 | 44.2 | 36.4 |
| | DCMEM | **84.3** | **75.7** | **68.0** | **51.3** | **37.2** | **28.0** | **79.9** | **68.5** | **60.3** |
| | MVAE | 28.2 | 13.1 | 7.9 | 27.0 | 12.0 | 7.3 | 30.5 | 17.4 | 10.5 |
| | MoPoE | 19.4 | 5.1 | 2.4 | 25.4 | 11.6 | 6.1 | 26.9 | 10.2 | 5.8 |
| 0.25 | MEME | 16.9 | 1.7 | 0.2 | 16.1 | 1.0 | 0.1 | 17.2 | 1.8 | 0.3 |
| | MVP | 41.7 | 30.1 | 20.4 | 20.6 | 6.5 | 4.2 | 31.4 | 20.9 | 17.7 |
| | DCMEM | **73.6** | **62.0** | **51.3** | **53.1** | **39.8** | **32.6** | **82.7** | **70.9** | **64.3** |

Table 6: Quantitative comparison of clustering performance based on latent representations under different missing rates in the Caption modality on the CUBICC dataset. Fraction indicates the proportion of paired samples relative to the full training set. Each model is trained on a dataset consisting of paired data at a proportion of Fraction and unimodal Image data at a proportion of 1-Fraction, and evaluated on the complete test set.

| Fraction | Methods | Image Representation | | | Caption Representation | | | Joint Representation | | |
|---|---|---|---|---|---|---|---|---|---|---|
| | | ACC | NMI | ARI | ACC | NMI | ARI | ACC | NMI | ARI |
| | MVAE | 29.0 | 12.1 | 7.5 | 22.6 | 11.3 | 3.8 | 33.7 | 26.9 | 15.1 |
| | MoPoE | 27.9 | 15.8 | 7.8 | 38.2 | 23.0 | 15.2 | 42.0 | 27.9 | 17.0 |
| 0.75 | MEME | 24.5 | 9.1 | 4.7 | 24.8 | 15.7 | 6.5 | 24.5 | 16.2 | 7.9 |
| | MVP | 58.8 | 48.2 | 37.2 | 42.9 | 30.1 | 21.6 | 52.3 | 50.5 | 36.7 |
| | DCMEM | **82.0** | **70.5** | **63.2** | **60.8** | **44.7** | **33.9** | **83.9** | **74.5** | **67.2** |
| | MVAE | 28.0 | 11.3 | 6.4 | 20.4 | 6.3 | 3.1 | 46.2 | 33.1 | 23.0 |
| | MoPoE | 27.8 | 13.1 | 8.0 | 33.0 | 19.3 | 12.0 | 31.0 | 18.9 | 11.6 |
| 0.5 | MEME | 17.0 | 1.3 | 0.3 | 16.7 | 1.4 | 0.2 | 17.1 | 1.6 | 0.3 |
| | MVP | 50.0 | 36.0 | 27.1 | 45.8 | 27.4 | 19.0 | 64.1 | 50.9 | 41.8 |
| | DCMEM | **81.0** | **68.4** | **61.9** | **57.2** | **38.8** | **29.5** | **82.0** | **70.1** | **63.4** |
| | MVAE | 29.5 | 17.3 | 8.9 | 21.2 | 6.0 | 2.9 | 35.9 | 24.6 | 15.7 |
| | MoPoE | 24.5 | 7.6 | 4.5 | 25.0 | 10.3 | 5.4 | 27.6 | 11.7 | 7.0 |
| 0.25 | MEME | 15.9 | 0.8 | 0.1 | 16.4 | 1.1 | 0.1 | 16.3 | 1.0 | 0.1 |
| | MVP | 51.1 | 33.4 | 25.8 | 40.1 | 24.1 | 14.7 | 54.5 | 41.8 | 31.1 |
| | DCMEM | **65.6** | **55.2** | **42.2** | **54.7** | **40.4** | **29.3** | **67.4** | **62.9** | **50.4** |

undesirably low distances in off-diagonal entries, which implies confusion between unrelated classes. Notably, only our model consistently achieves a strong diagonal with low intra-class distances and high inter-class distances across both datasets, highlighting its superior capability in capturing and preserving cross-modal semantic structure.

## C.5  Ablation Study

The explicit mathematical definitions of the objective functions used in the ablation study are as follows:

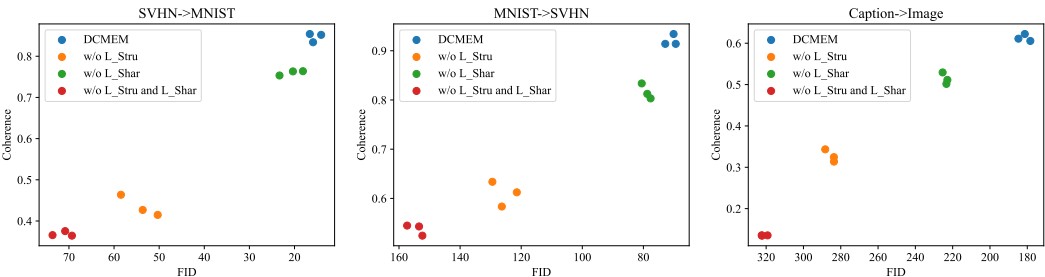

Figure 31: Generation performance with different modules ablated. $\mathcal{L}_{Stru}$ represents the structured representation learning module and $\mathcal{L}_{Shar}$ represents the shared representations alignment module.

Table 7: Clustering performance of joint latent representations with different modules ablated. $\mathcal{L}_{Stru}$ represents the structured representation learning module and $\mathcal{L}_{Shar}$ represents the shared representations alignment module.

| Datasets | | | MNIST_SVHN | | | CUBICC | | |
|---|---|---|---|---|---|---|---|---|
| $\mathcal{L}_{ELBO}$ | $\mathcal{L}_{Stru}$ | $\mathcal{L}_{Shar}$ | ACC | NMI | ARI | ACC | NMI | ARI |
| ✓ | | | 50.6±3.1 | 32.4±2.1 | 27.4±2.2 | 26.4±0.7 | 13.7±0.2 | 10.3±0.1 |
| ✓ | ✓ | | 91.5±2.4 | 80.6±1.8 | 82.0±1.2 | 84.6±1.8 | 74.1±0.9 | 68.6±2.1 |
| ✓ | | ✓ | 72.7±1.9 | 63.5±1.3 | 57.7±1.2 | 49.3±1.3 | 50.7±1.9 | 35.0±0.9 |
| ✓ | ✓ | ✓ | **99.5±0.1** | **98.4±0.2** | **98.9±0.1** | **86.3±1.8** | **76.8±2.8** | **71.5±3.1** |

(1) $\mathcal{L}_{\text{ELBO}}$ denotes the variational lower bound derived in Section 3.1. Under our mutual supervision setup, it includes both $s \to z \to t$ and $t \to z \to s$ directions. For the $s \to z \to t$ direction, the ELBO term is given by: $\mathcal{L}_{\text{ELBO}}^{s \to t} = \mathbb{E}_{q_\phi(z_s, w_s|s)} \left[ \log \frac{p_{\psi_z}(z_s|t)p(w_s)p_\theta(s|z_s, w_s)}{q_{\phi_z}(z_s|s)q_{\phi_w}(w_s|s)} \right] + \log q_{\phi_z, \phi}(t \mid s) + \log p(t)$. A symmetric term is used for the $t \to z \to s$ direction. Together, they form the total $\mathcal{L}_{\text{ELBO}}$ used in training.

(2) $\mathcal{L}_{\text{Stru}}$ corresponds to the Structured Representation Learning term introduced in Section 3.1. It also includes bidirectional modeling. For example, the $s \to z \to t$ direction includes a reconstruction term and a latent distribution alignment term: $\mathcal{L}_{\text{Stru}}^{s \to t} = \mathbb{E}_{q_\phi(\boldsymbol{z}_s, \boldsymbol{w}_s|s)p_D(\boldsymbol{s})} \log p_\theta(\boldsymbol{s} \mid \boldsymbol{z}_s, \boldsymbol{w}_s) - \mathbb{E}_{p_D(\boldsymbol{s}, \boldsymbol{t})} \left[ D_{KL} \left( q_{\phi_z} \left( \boldsymbol{z}_s = \boldsymbol{z} \mid \boldsymbol{s} \right) || q_{\psi_z} \left( \boldsymbol{z}_t = \boldsymbol{z} \mid \boldsymbol{t} \right) \right) \right]$ and vice versa for the $t \to z \to s$ direction.

(3) $\mathcal{L}_{\text{Shar}}$ corresponds to the Shared Representations Alignment term introduced in Section 3.1. It captures the mutual information between $z_s$ and $z_t$, defined as: $\mathcal{L}_{\text{Shar}} = \alpha I(z_s; z_t)$, where $I(\cdot; \cdot)$ is estimated via contrastive learning.

To evaluate the contribution of each component in our model, we perform an ablation study by selectively removing the structured representation learning ($\mathcal{L}_{Stru}$) and the shared representations alignment ($\mathcal{L}_{Shar}$). The results are summarized in Figure 31 and Table 7. As shown in Figure 31, we evaluate the contribution of each module to generation performance using FID and coherence scores across three tasks. The full model consistently achieves the lowest FID and highest coherence scores, indicating superior visual fidelity and semantic consistency. Removing either $\mathcal{L}_{Stru}$ or $\mathcal{L}_{Shar}$ leads to a clear decline in performance, confirming the necessity of both components. Table 7 reports the clustering performance of different ablation settings on MNIST-SVHN and CUBICC datasets. We observe that omitting $\mathcal{L}_{Shar}$ results in a modest performance drop as the model loses the alignment constraint for shared latent features, leading to suboptimal cross-modal representation. In contrast, removing $\mathcal{L}_{Stru}$ causes a significant decline in all metrics. This suggests that without proper disentanglement of shared and modality-specific information, the model fails to preserve meaningful semantic structure in the shared space. Overall, the ablation results demonstrate that both structured representation learning and shared representations alignment are indispensable for achieving strong performance in both generation and clustering tasks.

To further isolate our architectural contribution, we conduct experiments on the CUBICC dataset by enhancing MVP with the VampPrior mechanism. MVP is selected as the strongest non-VampPrior baseline in terms of cross-modal generation and its competitive performance in clustering and classification under various pairing rates (Figures 12, 23; Tables 5, 6). The resulting variant, MVP_VP,

Table 8: Generation performance of MVP_VP (The subscript of each metric indicates the observed modality. For example, Coherence_Image (Fraction = 0.25) denotes that the training data consists of 25% paired samples and 75% unimodal Image samples).

| Fraction | Methods | Coherence_Image | FID_Image | Coherence_Caption | FID_Caption |
|---|---|---|---|---|---|
| 0.75 | MVP | 0.242 | 253.243 | 0.239 | 250.773 |
| | MVP_VP | 0.241 | 265.464 | 0.234 | 251.384 |
| | DCMEM | **0.497** | **203.981** | **0.515** | **207.708** |
| 0.5 | MVP | 0.237 | 256.383 | 0.224 | 255.310 |
| | MVP_VP | 0.215 | 260.604 | 0.223 | 259.447 |
| | DCMEM | **0.517** | **204.286** | **0.471** | **212.218** |
| 0.25 | MVP | 0.231 | 259.064 | 0.195 | 256.498 |
| | MVP_VP | 0.217 | 250.562 | 0.187 | 271.769 |
| | DCMEM | **0.313** | **214.203** | **0.294** | **221.351** |

Table 9: Classification accuracy of MVP_VP (The subscript of each metric indicates the observed modality. For example, Image Representation_Image (Fraction = 0.25) denotes that the training data consists of 25% paired samples and 75% unimodal Image samples).

| Fraction | Methods | Image Representation_Image | Caption Representation_Image | Image Representation_Caption | Caption Representation_Caption |
|---|---|---|---|---|---|
| 0.75 | MVP | 0.864 | 0.561 | 0.877 | 0.566 |
| | MVP_VP | 0.824 | 0.512 | 0.804 | 0.532 |
| | DCMEM | **0.866** | **0.631** | **0.873** | **0.645** |
| 0.5 | MVP | 0.865 | 0.557 | 0.825 | 0.537 |
| | MVP_VP | 0.754 | 0.520 | 0.755 | 0.485 |
| | DCMEM | **0.871** | **0.618** | **0.879** | **0.620** |
| 0.25 | MVP | 0.702 | 0.518 | 0.676 | 0.458 |
| | MVP_VP | 0.635 | 0.453 | 0.647 | 0.386 |
| | DCMEM | **0.712** | **0.526** | **0.717** | **0.576** |

uses pseudo-points from the missing modality to construct a Gaussian mixture prior, which guides latent learning from the observed modality. As shown in the Tables 8, 9 and 10, MVP_VP does not yield consistent improvements over the original MVP baseline. On the contrary, it often leads to a degradation in generation metrics, classification accuracy and clustering performance, particularly at lower pairing rates. We hypothesize that this is due to a mismatch in modeling assumptions: MVP relies on cycle-consistency alignment, which degenerates to a trivial alignment (i.e., with itself) when only one modality is present, yielding zero loss for such cases. The introduction of VampPrior forces these unpaired samples to align with a prior constructed from the missing modality, introducing a non-trivial loss term that may disrupt the overall optimization, especially since the alignment does not follow the same cyclic mechanism as MVP's original design. Apart from these results, it is worth noting that two baseline models, MMVM and MEME , which use a similar VampPrior strategy, also underperform compared to our model. This further indicates that our performance gains stem not only from the use of VampPrior, but from the integration of disentangled representation learning and mutual information alignment within a unified mutual supervision framework, which ensures consistent robustness under both paired and missing data scenarios.

## C.6 Parameter Analysis

To evaluate the impact of the shared representations alignment component, we conduct a parameter analysis on its weighting factor $\alpha$. As illustrated in Figure 32, we plot the FID and Coherence scores under different values of $\alpha$ across three cross-modal generation tasks. The results reveal that the model achieves optimal performance when $\alpha$ is set to 0.5 or 1. In this range, the alignment module effectively bridges modality gaps by aligning latent representations, thereby preserving semantic consistency and enhancing generation quality. In contrast, when $\alpha$ is too small, the alignment term contributes minimally and results in suboptimal cross-modal coherence. On the other hand, excessively large $\alpha$ values may lead to overfitting or over-alignment which adversely affects performance. In addition, Table 11 reports the clustering performance of joint latent representations under different $\alpha$ values. We observe that the model maintains strong performance when $\alpha$ ranges from 0.1 to 1. However,

Table 10: Clustering accuracy of MVP_VP (The subscript of each metric indicates the observed modality. For example, Image Representation_Image (Fraction = 0.25) denotes that the training data consists of 25% paired samples and 75% unimodal Image samples).

| Fraction | Methods | Image Representation_Image | Caption Representation_Image | Joint Representation_Image |
|---|---|---|---|---|
| | MVP | 58.8 | 42.9 | 52.3 |
| 0.75 | MVP_VP | 29.4 | 32.2 | 34.5 |
| | DCMEM | **82.0** | **60.8** | **83.9** |
| | MVP | 50.0 | 45.8 | 64.1 |
| 0.5 | MVP_VP | 40.2 | 36.7 | 44.2 |
| | DCMEM | **81.0** | **57.2** | **82.0** |
| | MVP | 51.1 | 40.1 | 54.5 |
| 0.25 | MVP_VP | 25.0 | 25.4 | 27.8 |
| | DCMEM | **65.6** | **54.7** | **67.4** |

| Fraction | Methods | Image Representation_Caption | Caption Representation_Caption | Joint Representation_Caption |
|---|---|---|---|---|
| | MVP | 53.1 | 44.3 | 72.7 |
| 0.75 | MVP_VP | 41.9 | 32.3 | 39.6 |
| | DCMEM | **83.6** | **62.6** | **84.1** |
| | MVP | 58.7 | 31.0 | 52.4 |
| 0.5 | MVP_VP | 37.8 | 34.0 | 42.9 |
| | DCMEM | **84.3** | **51.3** | **79.9** |
| | MVP | 41.7 | 20.6 | 31.4 |
| 0.25 | MVP_VP | 24.8 | 19.9 | 22.7 |
| | DCMEM | **73.6** | **53.1** | **82.7** |

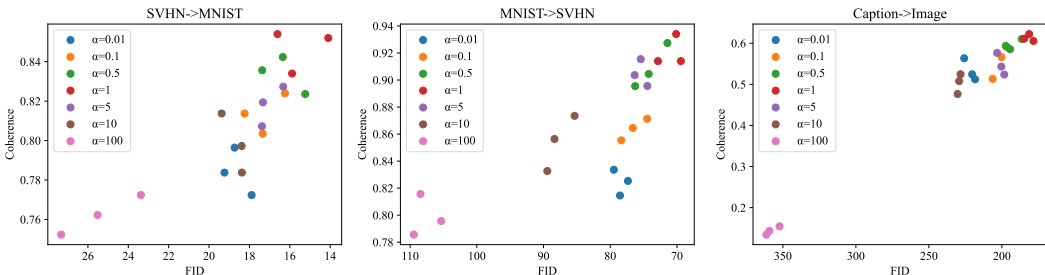

Figure 32: Generation performance under different $\alpha$ values.

Table 11: Clustering performance of joint latent representations under different $\alpha$ values.

| Datasets | MNIST_SVHN | | | CUBICC | | |
|---|---|---|---|---|---|---|
| $\alpha$ | ACC | NMI | ARI | ACC | NMI | ARI |
| 0.01 | 94.4±1.2 | 90.0±1.0 | 91.4±1.5 | 84.5±2.1 | 74.1±1.7 | 68.6±1.7 |
| 0.1 | 97.9±0.3 | 95.8±0.2 | 96.7±0.3 | **86.7±1.1** | 75.7±1.4 | 69.8±1.9 |
| 0.5 | 99.1±0.2 | **98.6±0.4** | 97.9±0.3 | 85.6±1.6 | 76.4±1.3 | 71.2±1.5 |
| 1 | **99.5±0.1** | 98.4±0.2 | **98.9±0.1** | 86.3±1.8 | **76.8±2.8** | **71.5±3.1** |
| 5 | 96.5±1.1 | 91.1±1.3 | 92.4±1.2 | 83.4±1.4 | 72.3±0.8 | 66.6±1.6 |
| 10 | 92.3±1.9 | 82.1±1.4 | 83.8±1.7 | 75.0±1.3 | 69.7±0.7 | 59.3±0.9 |
| 100 | 85.9±1.7 | 72.6±1.4 | 71.8±1.8 | 67.0±1.1 | 49.3±1.4 | 40.4±1.3 |

when $\alpha$ exceeds this range, the clustering performance degrade significantly, further confirming the importance of a well-balanced alignment strength. Based on both generation and clustering results, we recommend setting $\alpha$ between 0.5 and 1 in practice for robust and consistent performance.

## C.7 Computational Resources

All experiments are conducted on a machine equipped with an NVIDIA GeForce RTX 2080 Ti GPU and 64 GB of RAM. For the MNIST-SVHN dataset, each run uses 4 CPU workers and approximately 10 GB of GPU memory, with an average training time of around 8 hours per run. We evaluate 9 different methods, each with 3 random seeds, resulting in a total compute time of approximately 216 GPU hours ($8 \times 9 \times 3$). For the CUBICC dataset, each run uses 2 CPU workers and approximately 9 GB of GPU memory. Each training run takes about 16 hours on average. Evaluating 9 methods over 3 seeds results in a total compute time of roughly 432 GPU hours ($16 \times 9 \times 3$). For the Human Breast Cancer dataset, each run uses 2 CPU workers and around 4 GB of GPU memory. The average runtime is approximately 2 hours. With 14 methods and 3 seeds, the total compute time amounts to about 84 GPU hours ($2 \times 14 \times 3$). In total, the experiments require approximately 732 GPU hours. Additional GPU time is used during model development and hyperparameter tuning, which is not included in the above calculation.

