# OpenReview forum: "Disentangled Cross-Modal Representation Learning with Enhanced Mutual Supervision"
_NeurIPS.cc/2025/Conference — NeurIPS 2025 poster_

### Official Review · Reviewer_GUru · 2025-06-13

**Clarity:** 3
**Significance:** 4
**Originality:** 4
**Rating:** 5
**Confidence:** 5

**Summary:**

This paper proposes a disentangled mutual supervision framework for cross-modal representation learning with missing modalities. It introduces shared and modality-specific latent variables, guided by mutual information regularization, to learn semantically aligned and disentangled representations. Experiments on MNIST-SVHN, CUBICC, and a spatial transcriptomics dataset show that the method outperforms eight multimodal VAEs in generation quality, semantic consistency, and robustness, demonstrating its effectiveness across both vision and biomedical domains.

**Questions:**

1. The authors should provide more details on the network architecture and parameter settings used for the spatial transcriptomics dataset.
2. The implementation of $q_{\phi}(t | z_s)$ is unclear because it does not involve the variable$ w$; this raises questions about how this distribution is modeled in practice.
3. The authors are encouraged to discuss the numerical stability and computational method of the importance sampling weight $\frac{q(t|z_s)}{q(t|s)}$ in Equation (2). Additional explanation would help readers understand its validity and implementation.
4. The authors should provide the explicit mathematical formulations of the loss functions used in the ablation experiments would improve clarity and help readers better understand the individual contributions of each component.
5. Minor point: The number of classes for the MNIST-SVHN dataset is not clearly stated.

**Ethical Concerns:**

["NO or VERY MINOR ethics concerns only"]

**Final Justification:**

Thank the author for their rebuttal. All my concerns have been resolved, and I will maintain the score.

**Limitations:**

Yes.

**Paper Formatting Concerns:**

1. The notation contains formatting issues such as an incomplete term I ()(e.g., line 462), which should be corrected for clarity and correctness.
2. There are instances where words are not properly separated by spaces (e.g., line 517).
3. Some equations contain extraneous symbols, such as in Equation (18).

**Quality:**

4

**Strengths And Weaknesses:**

Strengths
1. The method introduces a unique mutual supervision mechanism between modality-specific and shared latent variables, promoting semantic alignment while preserving modality-specific features. This improves interpretability and robustness compared to adversarial or alignment-based methods.
2. The framework naturally supports partially observed inputs without requiring separate strategies for missing data, making it practical for real-world multimodal scenarios.
3. The model’s ability to reveal spatial patterns in real biological data demonstrates its practical utility and potential for biomedical discovery.

Weaknesses
1. The paper provides insufficient details on the network architecture and parameter settings used for the spatial transcriptomics dataset, which limits reproducibility and understanding of model design.
2. The modeling of $q_{\phi}(t | z_s)$ remains unclear due to the absence of the latent variable $w$, leaving ambiguity in how this conditional distribution is practically implemented.
3. The importance sampling weight $\frac{q(t|z_s)}{q(t|s)}$ introduced in Equation (2) lacks discussion regarding its numerical stability and computational method, which raises concerns about its practical validity and implementation.
4. The ablation study refers to loss functions without explicitly linking them to corresponding equations in the main text, reducing clarity and making it harder to interpret the individual contributions of each component.

---

> ### Author Rebuttal · Authors · 2025-07-31
>
> We thank the reviewer for the encouraging feedback and incisive questions.
>
> >**Q1, W1: The authors should provide more details on the network architecture and parameter settings used for the spatial transcriptomics dataset.**
>
> **A1**: For the spatial transcriptomics dataset, we preprocess the gene expression data by selecting the top 3000 highly variable genes, followed by standard normalization and log-transformation. Both the encoder and decoder for this modality are implemented as fully connected neural networks. For the tissue morphology modality, input images are resized to 128×128 pixels. To incorporate spatial context, we adopt the CoordConv (NeurIPS, 2018) technique by appending the 2D spatial coordinates (x, y) as two additional input channels, resulting in a 5-channel input. This modality is processed using convolutional neural networks for both encoding and decoding. Model hyperparameters are set as follows: the weighting coefficient $\alpha$ is fixed at 1 and both the shared and modality-specific latent dimensions are set to 32. Optimization is performed using Adam with a learning rate of 5e-4, a batch size of 64 and 100 training epochs. The detailed architectures are summarized below:
>
> Transcriptomics modality:
> |Encoder_z|Encoder_w|Decoder|
> |-|-|-|
> |$\text{Input} \in {\mathbb{R}^{3000}}$|$\text{Input} \in {\mathbb{R}^{3000}}$|$\text{Input} \in {\mathbb{R}^{64}}$|
> |FC. 512 & ReLU|FC. 512 & ReLU|FC. 512 & ReLU|
> |FC. 512 & ReLU|FC. 512 & ReLU|FC. 512 & ReLU|
> |FC. 32, FC. 32 & Softplus|FC. 32, FC. 32 & Softplus|FC. 3000|
>
> Morphology modality with spatial coordinates:
> |Encoder_z|Encoder_w|Decoder|
> |-|-|-|
> |$\text{Input} \in {\mathbb{R}^{5 \times 128 \times 128}}$|$\text{Input} \in {\mathbb{R}^{5 \times 128 \times 128}}$|$\text{Input} \in {\mathbb{R}^{64}}$|
> |$4 \times 4$ conv. 64 stride 2 pad 1 & ReLU|$4 \times 4$ conv. 64 stride 2 pad 1 & ReLU|$4 \times 4$ upconv. 512 stride 1 pad 0 & ReLU|
> |$4 \times 4$ conv. 128 stride 2 pad 1 & ReLU|$4 \times 4$ conv. 128 stride 2 pad 1 & ReLU|$4 \times 4$ upconv. 512 stride 2 pad 1 & ReLU|
> |$4 \times 4$ conv. 256 stride 2 pad 1 & ReLU|$4 \times 4$ conv. 256 stride 2 pad 1 & ReLU|$4 \times 4$ upconv. 256 stride 2 pad 1 & ReLU|
> |$4 \times 4$ conv. 512 stride 2 pad 1 & ReLU|$4 \times 4$ conv. 512 stride 2 pad 1 & ReLU|$4 \times 4$ upconv. 128 stride 2 pad 1 & ReLU|
> |$4 \times 4$ conv. 512 stride 2 pad 1 & ReLU|$4 \times 4$ conv. 512 stride 2 pad 1 & ReLU|$4 \times 4$ upconv. 64 stride 2 pad 1 & ReLU|
> |$4 \times 4$ conv. 32 stride 1 pad 0, $4 \times 4$ conv. 32 stride 1 pad 0 & Softplus|$4 \times 4$ conv. 32 stride 1 pad 0, $4 \times 4$ conv. 32 stride 1 pad 0 & Softplus|$4 \times 4$ upconv. 5 stride 2 pad 1 & Sigmoid|
>
> >**Q2, W2: The implementation of $q_\phi(t|z_s)$ is unclear because it does not involve the variable $w$; this raises questions about how this distribution is modeled in practice.**
>
> **A2**: In our framework, $q_\phi(t \mid z_s)$ serves a key role in the mutual supervision mechanism. Specifically, $z_s$ is the shared latent representation inferred from modality $s$ and $q_\phi(t \mid z_s)$ models the reconstruction of modality $t$ based solely on this shared information. This setup is intentionally designed to exclude the private latent variable $w$, since the goal is to assess what information is common and transferable across modalities.
> In practice, we implement $q_\phi(t \mid z_s)$ by setting $w = 0$ and passing the concatenation $[z_s, 0]$ into the decoder of modality $t$.
> This design offers two key advantages: (1) it avoids introducing an additional decoder by reusing $p_\phi(t \mid z_s, w)$ with $w$ set to zero, keeping the architecture compact; and (2) it promotes effective disentanglement, as reconstructing $t$ from $z_s$ alone forces $z_s$ to capture modality-invariant, shared information.
>
> >**Q3, W3: The authors are encouraged to discuss the numerical stability and computational method of the importance sampling weight $\frac{q_\varphi(t \mid z_s)}{q_{\phi_z, \varphi}(t \mid s)}$ in Equation (2). Additional explanation would help readers understand its validity and implementation.**
>
> **A3**: The importance sampling weight $\frac{q_\varphi(t \mid z_s)}{q_{\phi_z, \varphi}(t \mid s)}$ in Equation (2) plays a critical role in our training objective, but we acknowledge that its direct computation can introduce numerical instability due to high variance in gradient estimates. This arises because both the numerator and denominator are parameterized distributions that are learned during training, and their stochastic nature may result in noisy, unreliable updates when used in Monte Carlo estimation of the ELBO.
> To address this issue, we adopt a stop-gradient strategy to stabilize training without compromising the objective, following a rationale similar to that in prior work [1, 2]. Concretely, we prevent gradients from flowing through the  $\frac{q_\varphi(t \mid z_s)}{q_{\phi_z, \varphi}(t \mid s)}$, treating it as a fixed scalar during backpropagation:
> stop_gradient$ \left(\frac{q_\varphi(t \mid z_s)}{q_{\phi_z, \varphi}(t \mid s)}\right)$.
> This modification ensures that the parameters are optimized based on more stable signals, as it avoids amplifying gradient noise through the ratio. Importantly, this treatment does not alter the forward computation of the objective but improves the robustness and reliability of the training dynamics.
> In summary, this design provides a practical and effective solution to variance-induced instability in training, while still aligning with the theoretical intent of the original ELBO formulation.
>
> >**Q4, W4: The authors should provide the explicit mathematical formulations of the loss functions used in the ablation experiments would improve clarity and help readers better understand the individual contributions of each component.**
>
> **A4**: The explicit mathematical definitions of the objective functions used in the ablation study are as follows：
>
>  (1) $\mathcal{L}\_{\text{ELBO}}$ denotes the variational lower bound derived in **Section 3.1**. Under our mutual supervision setup, it includes both $s \rightarrow z \rightarrow t$ and $t \rightarrow z \rightarrow s$ directions. For the $s \rightarrow z \rightarrow t$ direction, the ELBO term is given by: $\mathcal{L}\_{\text{ELBO}}^{s \rightarrow t} = \mathbb{E}\_{q_{\phi}(z_s, w_s \mid s)} \left[\log \frac{p_{\psi_z}(z_s \mid t) p(w_s) p_\theta(s \mid z_s, w_s)}{
> q_{\phi_z}(z_s \mid s) q_{\phi_w}(w_s \mid s)} \right]+ \log q_{\phi_z,\phi}(t \mid s) + \log p(t)$. A symmetric term is used for the $t \rightarrow z \rightarrow s$ direction. Together, they form the total $\mathcal{L}\_{\text{ELBO}}$ used in training.
>
> (2) $\mathcal{L}\_{\text{Stru}}$ corresponds to the Structured Representation Learning term introduced in **Section 3.1**. It also includes bidirectional modeling. For example, the $s \rightarrow z \rightarrow t$ direction includes a reconstruction term and a latent distribution alignment term: $\mathcal{L}\_{\text{Stru}}^{s \rightarrow t} = \mathbb{E}\_{q_\phi(z_s, w_s \mid s)} \left[\log p_\theta(s \mid z_s, w_s)\right] - D_{\text{KL}}\left( q_{\phi_z}(z_s \mid s) \|\| q_{\psi_z}(z_t \mid t) \right)$ and vice versa for the $t \rightarrow z \rightarrow s$ direction.
>
> (3) $\mathcal{L}\_{\text{Shar}}$ corresponds to the Shared Representations Alignment term introduced in **Section 3.1**. It captures the mutual information between $z_s$ and $z_t$, defined as: $\mathcal{L}\_{\text{Shar}} = \alpha I(z_s; z_t),$ where $I(\cdot;\cdot)$ is estimated via contrastive learning.
> For clarity, we will explicitly include the mathematical definitions of these objective terms in the appendix and cross-reference them in the ablation discussion.
>
> >**Q5: Minor point: The number of classes for the MNIST-SVHN dataset is not clearly stated.**
>
> **A5**: We clarify that the MNIST-SVHN dataset used in our experiments consists of 10 classes, corresponding to the digits 0 through 9. We will explicitly state this in the revised version to avoid any ambiguity.
>
> >**Paper Formatting Concerns: The notation contains formatting issues such as an incomplete term I ()(e.g., line 462), which should be corrected for clarity and correctness; There are instances where words are not properly separated by spaces (e.g., line 517); Some equations contain extraneous symbols, such as in Equation (18).**
>
> **A6**: We have carefully revised the manuscript to correct the issues mentioned: (1) The incomplete mutual information term $I()$  on line 462 has been fixed to correct mathematical expression $I(s; t; z_s)$. (2) All instances of missing spaces (e.g., line 517) have been reviewed and corrected to ensure proper word separation. (3) Extraneous or unintended symbols in Equation (18) and other equations have been removed or corrected for clarity and consistency.
>
> [1] Joy T, Schmon S M, Torr P H S, et al. Capturing label characteristics in {vae}s. In International Conference on Learning Representations, 2021.
>
> [2] Joy T, Shi Y, Torr P, et al. Learning Multimodal VAEs through Mutual Supervision. In International Conference on Learning Representations, 2022.

---

> > ### Comment · Reviewer_GUru · 2025-08-05
> >
> > Thank the author for their rebuttal. All my concerns have been resolved, and I will maintain the score.

---

> > > ### Author Response · Authors · 2025-08-05
> > >
> > > Thank you once again for your constructive and thoughtful comments. We're delighted to know that our revisions have effectively addressed your concerns.

---

### Official Review · Reviewer_RUfA · 2025-06-20

**Clarity:** 3
**Significance:** 4
**Originality:** 4
**Rating:** 5
**Confidence:** 4

**Summary:**

This paper proposes DCMEM, a novel variational framework for cross-modal representation learning that jointly disentangles shared and modality-specific latent variables using an information-theoretic objective. The model introduces a mutual supervision mechanism that aligns shared representations across modalities through cross-modal inference paths, avoiding restrictive assumptions used in prior work like PoE or MoE. An information bottleneck constraint is applied to promote semantic disentanglement, while mutual information regularization ensures consistency between the shared latent codes of different modalities. Notably, the model is designed to support training on partially observed multimodal data via an empirical prior approximation. Extensive experiments on MNIST-SVHN, CUBICC, and a spatial transcriptomics dataset demonstrate strong performance in generation, clustering, and classification tasks, particularly under high missing-view rates, validating both the flexibility and effectiveness of the proposed method.

**Questions:**

1. How is $I(z_s; z_t)$ approximated? What contrastive loss and projection architecture are used? What temperature or batch strategies are applied?

2. In Table 1, DCMEM underperforms CMVAE on the caption modality. Can you offer an explanation? Is this due to weaker textual representations, or modality imbalance?

3. How is the prior anchor set $\{u_i^t\}$ selected? Is it fixed across epochs or sampled per batch?

4. The initialization of K-means has not been clearly explained, such as whether random seed settings, running times, etc. have an impact on the results?

**Ethical Concerns:**

["NO or VERY MINOR ethics concerns only"]

**Final Justification:**

Thank the authors for their efforts in clarifying these points. As a result, the main issues have been addressed, and I will maintain my rating.

**Limitations:**

The authors explicitly state in the appendix that the model is specifically designed for bimodal data scenarios and may require non-trivial modifications to generalize to datasets with more than two modalities. This is a reasonable and appropriate limitation to acknowledge. However, this information currently appears only in the appendix, and would be more visible and useful if summarized briefly in the main paper, for instance in the conclusion. Additionally, while the authors mention potential positive societal impact in biomedical domains, the discussion would be strengthened by also addressing data privacy, fairness or ethical considerations related to patient-derived spatial transcriptomics data.

Suggestions:

1. Move or repeat the bimodal-design limitation in the main body of the paper.

2. Add a short statement discussing privacy protection and ethical compliance when working with biomedical data.

**Paper Formatting Concerns:**

1. Figures lack clear and complete captions: Several figures are presented with minimal or vague captions, without describing the experimental setup, meaning of colors or classes, or takeaway insights. NeurIPS guidelines require figure captions to be self-contained and explanatory.

2. Appendix lacks section clarity: The appendix contains useful derivations and experimental details but is not clearly structured. Long blocks of text mix derivations (Appendix A), implementation (Appendix B), and results (Appendix C) without clear separation or consistent heading styles.

**Quality:**

4

**Strengths And Weaknesses:**

Strengths

1. The proposed DCMEM framework effectively combines bi-directional mutual supervision ($ s \rightarrow z \rightarrow t $ and $ t \rightarrow z \rightarrow s $), shared-private latent disentanglement, and mutual information regularization into a coherent and principled architecture.

2. The model is capable of learning from partially paired multimodal data using an empirical prior approximation for missing modalities.

3. The method is extensively evaluated on three representative datasets spanning different domains (image-image, image-text, biomedical), and across multiple tasks including cross-modal generation, clustering, and classification.

4. DCMEM achieves state-of-the-art or highly competitive results compared to recent multimodal VAEs, particularly in clustering and generation tasks.

Weaknesses

1. On certain tasks such as the caption modality clustering in CUBICC, DCMEM underperforms CMVAE (Table 1), but no explanation is provided.

2. The paper mentions contrastive learning to estimate $I(z_s; z_t)$ but gives no details about the architecture or training strategy used for the contrastive term.

3. Key implementation choices are insufficiently specified. For example, it is unclear whether the prior anchor set ${u_i^t}$ is fixed or sampled per batch, and how K-means initialization (e.g., random seeds, number of runs) might affect the results.

4. While technically sound, some mathematical derivations (e.g., Eq. 2, 6) are presented with dense notation and limited intuition, reducing accessibility to a broader audience.

---

> ### Author Rebuttal · Authors · 2025-07-31
>
> We thank the reviewer for the encouraging feedback and valuable comments.
>
> >**Q1, W2: How is $I(z_s;z_t)$ approximated? What contrastive loss and projection architecture are used? What temperature or batch strategies are applied?**
>
> **A1:** The mutual information $I(z_s;z_t)$ between the shared latent variables $z_s$ and $z_t$ is approximated using a contrastive loss that encourages alignment between representations of paired inputs while distinguishing those from non-paired ones [1]. Given a batch of $B$ paired latent features $(h_s, h_t)$ sampled from $z_s$ and $z_t$, we concatenate them into a set of $2B$ vectors. Pairwise cosine similarities are computed and scaled by a fixed temperature $\tau = 0.5$ to construct the contrastive logits. Positive pairs are defined between the $i$-th feature in $h_s$ and the $i$-th feature in $h_t$, as they correspond to the paired input instances. All other pairs in the batch are treated as negatives. We apply a cross-entropy loss to encourage the model to assign higher similarity to positives than to negatives. We do not use any additional projection architecture. The batch size matches that of the main training process to ensure sufficient negative diversity.
>
> >**Q2, W1: In Table 1, DCMEM underperforms CMVAE on the caption modality. Can you offer an explanation? Is this due to weaker textual representations, or modality imbalance?**
>
> **A2:** The caption modality in the CUBICC dataset presents inherent challenges for clustering, as the captions are typically short and offer only high-level visual descriptions (e.g., “the bird has a black belly that turns into an orange breast with black wings”), which contain limited discriminative information for species-level categorization. Given that the ground-truth labels are defined based on image content, the caption modality is relatively weak in clustering signal. While this makes it difficult for any method to achieve high performance, our slightly lower NMI compared to CMVAE (ICLR, 2024) may reflect the limitations of our current textual representation learning. Nonetheless, DCMEM still achieves better ACC and ARI scores, suggesting it captures substantial structure from this modality despite the challenge.
>
> >**Q3, W3: How is the prior anchor set selected? Is it fixed across epochs or sampled per batch?**
>
> **A3:** The prior anchor set ${u^t}$ is dynamically sampled on a per-batch basis rather than being fixed throughout training. For each batch, we randomly select a set of samples from the $t$ modality, with the number of selected anchors matching the batch size. These samples are passed through the encoder to obtain a set of latent Gaussian distributions, which serve as components of a Gaussian Mixture Model (GMM), with uniform weights assigned to all components. Since the anchor set is resampled at each training step, the prior remains adaptive and batch-dependent. This dynamic design enables the model to better capture the evolving latent structure during training, improving both stability and expressiveness compared to a fixed prior.
>
> >**Q4: The initialization of K-means has not been clearly explained, such as whether random seed settings, running times, etc. have an impact on the results?**
>
> **A4:** For all methods, including ours and the baselines, we use the standard KMeans implementation from the scikit-learn library with the random seed fixed to 0 via the random_state parameter. This ensures consistent initialization across runs. As a result, the clustering outcomes are solely influenced by the learned latent representations rather than stochastic variations from KMeans initialization or multiple runs.
>
> >**W4: While technically sound, some mathematical derivations (e.g., Eq. 2, 6) are presented with dense notation and limited intuition, reducing accessibility to a broader audience.**
>
> **A5:** We will revise the manuscript to simplify the notation and provide more intuitive explanations for the key mathematical derivations (e.g., Eqs. 2 and 6) to improve clarity and accessibility for a broader audience.
>
> [1] van den Oord A, Li Y, Vinyals O. Representation learning with contrastive predictive coding. arXiv preprint arXiv:1807.03748, 2018.

---

### Official Review · Reviewer_Qzph · 2025-06-30

**Clarity:** 2
**Significance:** 2
**Originality:** 2
**Rating:** 4
**Confidence:** 4

**Summary:**

This paper proposes a variational autoencoder-based framework for cross-modal representation learning that explicitly disentangles shared and modality-specific factors using enhanced mutual supervision and information bottleneck constraints. The method ensures semantic coherence across modalities by enforcing mutual supervision between their shared latent representations, while also promoting clear disentanglement between shared and specific latent variables.

**Questions:**

1. The datasets used in this paper (such as MNIST-SVHN and CUBICC) are standard but relatively simple benchmarks in cross-modal research. Is the method proposed in the article applicable to complex datasets such as video-text and audio-visual?
2. The core contributions appear primarily as combinations or incremental refinements of existing methods. Could the authors clarify explicitly why the proposed combination of existing techniques is theoretically superior or unique compared to closely related existing methods? Can additional theoretical analyses or experiments be provided to demonstrate these unique advantages?
3. While the method claims to be robust to partial modality availability, it's unclear how performance scales under severe missingness or when one modality is significantly more informative than the other. How does the model perform when one modality is systematically less informative, noisy, or entirely missing?

**Ethical Concerns:**

["NO or VERY MINOR ethics concerns only"]

**Final Justification:**

Thanks for the authors' response, most of my concerns have been addressed. Thus, I will increase my rating to borderline accept.

**Limitations:**

Yes, the authors have acknowledged key limitations in Appendix A.4

**Paper Formatting Concerns:**

This paper follows the formatting instructions.

**Quality:**

2

**Strengths And Weaknesses:**

Strengths
1. The proposed model is technically sound, incorporating well-established theoretical concepts like mutual information regularization, disentanglement, and variational inference frameworks.
2. The paper is generally well-structured, with a logical flow from motivation through methodology to evaluation.
3. The proposed method addresses important challenges in multimodal representation learning, notably modality disentanglement and handling incomplete multimodal data

Weakness
1. The experimental settings rely on commonly used yet simplified datasets. The lack of evaluations on more challenging, realistic multimodal tasks (such as video-audio, complex textual descriptions, or noisy sensor data) reduces the paper’s broader practical significance.
2. The core technical contributions largely consist of recombinations and incremental refinements of existing techniques from prior literature. Although presented clearly, the paper does not introduce significantly new theoretical insights, nor does it thoroughly articulate why this particular combination of existing methods provides a critical advantage.
3. The mathematical exposition, especially concerning mutual information approximations and ELBO derivations, is densely presented and may impede broader understanding. Providing more intuitive explanations, intermediate interpretations, or summarizing key equations clearly would significantly enhance readability.

---

> ### Author Rebuttal · Authors · 2025-07-31
>
> Thank you for your valuable and insightful comments.
>
> >**Q1, W1: The datasets used in this paper (such as MNIST-SVHN and CUBICC) are standard but relatively simple benchmarks in cross-modal research. Is the method proposed in the article applicable to complex datasets such as video-text and audio-visual?**
>
> **A1:** First, we would like to clarify that the primary goal of our work is not to propose a universal solution applicable to all cross-modal scenarios (e.g., video-text, audio-visual), but rather to address critical and long-standing limitations in existing VAE-based cross-modal frameworks -- insufficient alignment of the shared latent variables across modalities and inadequate disentanglement between shared and private latents.
>
> To address this, we design a novel model architecture with explicit shared latent variables $z$ and modality-specific variables $w$ and leverages mutual information terms to regulate their contents. By maximizing the mutual information between $z$ and observed multimodal data while minimizing the redundancy between $z$ and $w$, our model achieves principled disentanglement. Furthermore, mutual-supervision-based learning scheme and enhanced alignment mechanisms promote better cross-modal consistency in the shared latent space, which directly benefits downstream tasks such as cross-modal generation, classification and clustering.
>
> We emphasize that experiments on standard benchmarks such as MNIST-SVHN and CUBICC are well-aligned with our technical objectives and previous studies in this field: these datasets offer clear structural contrasts between modalities (e.g., digits-house numbers, images-captions) and provide a controlled setting to rigorously test the ability of VAE-based models to disentangle and align multimodal latent spaces. Rather than covering all modality types, our focus is on enhancing the VAE-based framework and validating it within the well-established image-text setting. While extending our method to more complex modalities such as video or audio is in principle possible, it represents a distinct research direction. These modalities typically require specialized backbone architectures and feature extractors to capture temporal dependencies, which go beyond the scope of the current study.
>
> >**Q2, W2: The core contributions appear primarily as combinations or incremental refinements of existing methods. Could the authors clarify explicitly why the proposed combination of existing techniques is theoretically superior or unique compared to closely related existing methods? Can additional theoretical analyses or experiments be provided to demonstrate these unique advantages?**
>
> **A2:** Our model is not just a simple combination of existing components, but a principled and structured innovation built around a unified evidence lower bound (ELBO) that serves as the core training objective. This ELBO is specifically designed to address two core limitations in current VAE-based cross-modal models: the insufficient disentanglement between shared and modality-specific information, and insufficient alignment of the shared latent variables, both of which critically undermine cross-modal generation and representation quality.
>
> The motivation for addressing the insufficient disentanglement between shared and modality-specific information in our work arises from the observation that both explicit (e.g., MMVAE+ (ICLR, 2023), CMVAE (ICLR, 2024)) and implicit (e.g., MoPoE (ICLR, 2021), MMVM (NeurIPS, 2024)) disentanglement strategies in existing VAE-based methods often fail to prevent information leakage between shared and private representations **(Figures 29-30)**. For instance, MMVAE+ and CMVAE introduce a modality-specific latent space $w$ to encourage disentanglement but lack sufficient constraints. As **Section 4.2 (Figures 6-7)**, class-relevant shared information is still encoded in the private variables, leading to significant representation redundancy and degradation in generation and clustering performance.
>
> To overcome this, we explicitly define both shared ($z$) and private ($w$) latent variables, and introduce a structural mutual information objective to enforce effective disentanglement: (1) By maximizing the mutual information between the shared latent $z$ and observed modalities, we encourage $z$ to encode information that is commonly present across modalities. (2) By minimizing the mutual information between $z$ and $w$, we reduce redundancy and prevent leakage of shared information into the modality-specific space. This structured objective enables us to learn high-quality, disentangled latent representations, as evidenced by our ablation study **(Section B.7, Figure 31, Table 7)**. Removing the structural mutual information objective (w/o \$\mathcal{L}\_{\text{Stru}}\$) leads to a marked drop in generation and clustering performance, as $z$ fails to capture shared information while $w$ dominates the representation. This demonstrates that our improvements stem not from simple component aggregation, but from a theoretically grounded disentanglement mechanism that effectively separates shared and private factors.
>
> Moreover, we observe that in previous mutual supervision settings (e.g., MEME (ICLR, 2022)), shared representations inferred from different modalities are often not well-aligned in latent space **(Figures 10-11, 26-27)**. Our model directly aligns the shared representations via mutual information maximization and  encourages consistent shared representation inference across modalities at the sample level, which further enhances cross-modal generation and representation learning. As shown in **Section 4.3 (Figures 9-11)**, our model achieves superior modality alignment, while other VAE-based models (e.g., MVAE (NeurIPS, 2018), MoPoE, MMVM) fail to do so. Furthermore, our ablation study **(Section B.7, Table 7)** confirms the critical role of this alignment mechanism: removing the objective (w/o $\mathcal{L}_{\text{Shar}}$) leads to a noticeable performance drop in joint clustering tasks due to insufficient alignment of the shared latent representations across modalities.
>
> Finally, our model introduces a novel disentanglement structure (shared $z$, private $w$) combined with mutual information-based objectives, leading to substantial improvements in downstream tasks including cross-modal generation, classification, and clustering **(Sections 4.2-4.4, B.3-B.6)**. Taken together, these contributions demonstrate that our method is not a mere recombination of prior techniques, but a theoretically motivated and empirically validated framework that effectively addresses long-standing challenges in cross-modal VAE-based learning. We hope this clarification helps convey the novelty and significance of our work.
>
> >**Q3: While the method claims to be robust to partial modality availability, it's unclear how performance scales under severe missingness or when one modality is significantly more informative than the other. How does the model perform when one modality is systematically less informative, noisy, or entirely missing?**
>
> **A3:** In the original manuscript, we have reported the results under varying missing rates, including experiments with 75% modality missingness, which is considered a severe level of incompleteness. As reported in **Sections 4.3, B.3-B.4** of our paper **(Figures 5,12-15, 21-24, and Tables 3-6)**, our evaluations span cross-modal generation, classification and clustering tasks, consistently demonstrating the model's robustness under varying levels of modality missingness.
>
> To further address the reviewer's concern, we conduct additional experiments on the CUBICC dataset simulating an even more challenging scenario: 90% of the data contains only the Caption modality, and only 10% is paired data, meaning the Image modality is almost entirely missing during training. This setup effectively mimics a situation where modalities are both highly imbalanced and one modality is significantly less available or informative. As shown below, DCMEM still outperforms other baselines across generation, classification, and clustering tasks.
>
> Cross-Modal Generation:
> |Methods|Coherence↑|FID↓|
> |-|-|-|
> |MVAE|0.130|353.2|
> |MoPoE|0.140|291.8|
> |MEME|0.133|309.4|
> |MVP|0.189|276.9|
> |DCMEM|**0.290**|**239.6**|
>
> Classification Accuracy:
> |Methods|Image Representation|Caption Representation|
> |-|-|-|
> |MVAE|0.513|0.198|
> |MoPoE|0.384|0.313|
> |MEME|0.135|0.145|
> |MVP|0.674|0.353|
> |DCMEM|**0.706**|**0.559**|
>
> Joint Representation Clustering Performance:
> |Methods|ACC|NMI|ARI|
> |-|-|-|-|
> |MVAE|28.2|10.3|6.7|
> |MoPoE|18.1|2.3|0.9|
> |MEME|16.3|1.5|0.3|
> |MVP|30.8|15.4|10.9|
> |DCMEM|**66.8**|**55.5**|**44.8**|
>
> These results provide strong evidence that our model remains robust even when one modality is noisy, uninformative, or mostly missing. This robustness stems from DCMEM's ability to effectively leverage limited supervision and unimodal data, enabled by its disentangled latent structure, mutual information-guided alignment and mutual supervision mechanism. As discussed in the **Limitations section**, our framework assumes access to partially observed modalities; if one modality is entirely missing, the problem reduces to a unimodal setting, which falls outside the intended scope of our cross-modal integration design.
>
> >**W3: The mathematical exposition, especially concerning mutual information approximations and ELBO derivations, is densely presented and may impede broader understanding. Providing more intuitive explanations, intermediate interpretations, or summarizing key equations clearly would significantly enhance readability.**
>
> **A4:** To balance the rigor of the paper with space constraints, we have placed the detailed intermediate derivations in the appendix for readers who wish to explore them in depth. Meanwhile, we will add more intuitive explanations and summaries of key equations in the main text to improve the overall readability and accessibility of the paper.

---

> > ### Comment · Reviewer_Qzph · 2025-08-02
> > **Replay for Submission28220 by Reviewer Qzph**
> >
> > Thanks for the authors' response, most of my concerns have been addressed. Thus, I will increase my rating to borderline accept.

---

> > > ### Author Response · Authors · 2025-08-02
> > >
> > > We are encouraged that your concerns have been addressed. Thank you again for your valuable and insightful feedback!

---

### Official Review · Reviewer_iAZE · 2025-07-03

**Clarity:** 3
**Significance:** 3
**Originality:** 2
**Rating:** 5
**Confidence:** 3

**Summary:**

This paper presents DCMEM, a VAE-based framework for cross-modal representation learning. The model disentangles latent factors into shared and private components, using mutual supervision and the information bottleneck principle for robust alignment. A key feature is its ability to train on both complete and partial data, showing strong performance across generation, clustering, and classification tasks.

**Questions:**

1. The model uses both KL divergence and a contrastive MI estimator to align shared latents. Why are both mechanisms necessary, and what unique aspects of alignment does the contrastive term capture?
2. To isolate model's architectural contributions, has the author considered applying the same VampPrior mechanism to the strongest baseline of the missing data experiment?
3. Regarding the empirical prior for missing data, are the N anchors fixed pseudo-inputs, or are they sampled dynamically during training?
4. What is the intuition for why the model, with a simple coordinate encoding, outperforms specialized graph neural network models on the spatial transcriptomics task?

**Ethical Concerns:**

["NO or VERY MINOR ethics concerns only"]

**Final Justification:**

The new experimental results and clarifications have successfully addressed my concerns. Consequently, I believe the paper is now a clear 'Accept' and I have updated my score accordingly.

**Limitations:**

Yes, the authors have adequately addressed the limitations of their work in Appendix A.4. They explicitly state that the model is designed for bimodal data scenarios, which is a fair acknowledgment of the current scope.

**Quality:**

3

**Strengths And Weaknesses:**

Strengths:
- The comprehensive experimental validation across multiple tasks, diverse datasets, and various data missingness rates provides strong empirical evidence for the method's effectiveness. This thoroughness makes a compelling case for the model's performance.
- The model's native ability to handle partially observed data is a significant practical advantage, supported by a valid ELBO derivation. This feature enhances its real-world applicability where complete data pairs are often unavailable.

Weaknesses:
- The method for handling missing data relies on an empirical prior over N anchors, but the paper fails to specify how these anchors are selected or updated.
- The comparison on incomplete data may not be entirely fair, as the proposed model leverages a VampPrior-style mechanism that some baselines lack. The observed performance gains might stem more from this specific prior than from the core architectural innovations of DCMEM.
- The model claims to disentangle shared and private variables, but the shared latent z is inferred from a single modality before being regularized. This indirect enforcement of "sharedness" may be less effective at purging modality-specific attributes than methods that infer the shared latent from a joint posterior.

---

> ### Author Rebuttal · Authors · 2025-07-31
>
> We thank the reviewer for the encouraging feedback and helpful suggestions.
>
> >**Q1: The model uses both KL divergence and a contrastive MI estimator to align shared latents. Why are both mechanisms necessary and what unique aspects of alignment does the contrastive term capture?**
>
> **A1:** While both the KL divergence and the contrastive mutual information (MI) estimator align the shared latent representations, they serve complementary purposes. The KL divergence term $D_{KL}(q_{\phi_z}(z_s = z \mid s) \|\| q_{\psi_z}(z_t = z \mid t))$ arises from a variational upper bound on $I(z_s; s \mid t)$, which measures the view-specific information retained in the shared latent variable $z_s$ beyond what is predictable from the other modality $t$. Minimizing this term encourages $z_s$ to discard modality-specific redundancy and retain only information accessible to both modalities. In this way, the KL divergence provides a global, distribution-level regularization that promotes the extraction of shared, modality-invariant features. However, KL divergence alone cannot guarantee that the shared latents from both modalities are semantically aligned for individual samples. The contrastive MI estimator directly maximizes $I(z_s, z_t)$, explicitly pulling together corresponding samples (positives) across modalities and pushing apart mismatched pairs (negatives). This provides instance-level alignment and enforces semantic consistency. Using both terms ensures that the shared latent variables are aligned globally in distribution and locally in semantics. Removing either term would weaken the disentanglement and cross-modal consistency, as each addresses a distinct aspect of the alignment problem.
>
> >**Q2, W2: To isolate model's architectural contributions, has the author considered applying the same VampPrior mechanism to the strongest baseline of the missing data experiment?**
>
> **A2:** To isolate our architectural contribution, we conduct experiments on the CUBICC dataset by enhancing MVP with the VampPrior mechanism. MVP (ICLR, 2025) is selected as the strongest non-VampPrior baseline in terms of cross-modal generation and its competitive performance in clustering and classification under various pairing rates **(Figures 12, 23; Tables 5, 6)**. The resulting variant, MVP_VP, uses pseudo-points from the missing modality to construct a Gaussian mixture prior, which guides latent learning from the observed modality.
>
> As shown in the tables below, MVP_VP does not yield consistent improvements over the original MVP baseline. On the contrary, it often leads to a degradation in generation metrics, classification accuracy and clustering performance, particularly at lower pairing rates. We hypothesize that this is due to a mismatch in modeling assumptions: MVP relies on cycle-consistency alignment, which degenerates to a trivial alignment (i.e., with itself) when only one modality is present, yielding zero loss for such cases. The introduction of VampPrior forces these unpaired samples to align with a prior constructed from the missing modality, introducing a non-trivial loss term that may disrupt the overall optimization, especially since the alignment does not follow the same cyclic mechanism as MVP's original design.
>
> Generation Performance (The subscript of each metric indicates the observed modality. For example, Coherence_Image (Fraction=0.25) denotes that the training data consists of 25% paired samples and 75% unimodal Image samples.):
> |Fraction|Methods|Coherence_Image↑|FID_Image↓|Coherence_Caption↑|FID_Caption↓|
> |-|-|-|-|-|-|
> |0.75|MVP|0.242|253.243|0.239|250.773|
> ||MVP_VP|0.241|265.464|0.234|251.384|
> ||DCMEM|**0.497**|**203.981**|**0.515**|**207.708**|
> |0.5|MVP|0.237|256.383|0.224|255.310|
> ||MVP_VP|0.215|260.6037|0.223|259.447|
> ||DCMEM|**0.517**|**204.286**|**0.471**|**212.218**|
> |0.25|MVP|0.231|259.064|0.195|256.498|
> ||MVP_VP|0.217|250.562|0.187|271.769|
> ||DCMEM|**0.313**|**214.203**|**0.294**|**221.351**|
>
> Classification Accuracy:
> |Fraction |Methods |Image Representation_Image |Caption Representation_Image |Image Representation_Caption |Caption Representation_Caption |
> |-|-|-|-|-|-|
> |0.75|MVP|0.864|0.561|**0.877**|0.566|
> ||MVP_VP|0.824|0.512|0.804|0.532|
> ||DCMEM|**0.866**|**0.631**|0.873|**0.645**|
> |0.5|MVP|0.865|0.557|0.825|0.537|
> ||MVP_VP|0.754|0.520|0.755|0.485|
> ||DCMEM|**0.871**|**0.618**|**0.879**|**0.620**|
> |0.25|MVP|0.702|0.518|0.676|0.458|
> ||MVP_VP|0.635|0.453|0.647|0.386|
> ||DCMEM|**0.712**|**0.526**|**0.717**|**0.576**|
>
> Clustering Accuracy:
> |Fraction |Methods |Image Representation_Image |Caption Representation_Image |Joint Representation_Image |Image Representation_Caption |Caption Representation_Caption |Joint Representation_Caption |
> |-|-|-|-|-|-|-|-|
> |0.75|MVP|58.8|42.9|52.3|53.1|44.3|72.7|
> ||MVP_VP|29.4|32.2 |34.5|41.9 |32.3 |39.6 |
> ||DCMEM|**82.0**|**60.8**|**83.9**|**83.6**|**62.6**|**84.1**|
> |0.5|MVP|50.0|45.8 |64.1|58.7|31.0 |52.4|
> ||MVP_VP|40.2|36.7 |44.2|37.8 |34.0 |42.9 |
> ||DCMEM|**81.0**|**57.2**|**82.0**|**84.3**|**51.3**|**79.9**|
> |0.25|MVP|51.1|40.1 |54.5|41.7 |20.6 |31.4 |
> ||MVP_VP|25.0|25.4 |27.8|24.8 |19.9 |22.7|
> ||DCMEM|**65.6**|**54.7**|**67.4**|**73.6**|**53.1**|**82.7**|
>
> Apart from the above results, it is worth noting that two baseline models, MMVM (NeurIPS, 2024) and MEME (ICLR, 2022), which use a similar VampPrior strategy, also underperform compared to our model. This further indicates that our performance gains stem not only from the use of VampPrior, but from the integration of disentangled representation learning and mutual information alignment within a unified mutual supervision framework, which ensures consistent robustness under both paired and missing data scenarios.
>
> >**Q3, W1: Regarding the empirical prior for missing data, are the N anchors fixed pseudo-inputs, or are they sampled dynamically during training?**
>
> **A3:** The prior anchor set ${u^t}$ is dynamically sampled on a per-batch basis rather than being fixed throughout training. For each batch, we randomly select a set of samples from the $t$ modality, with the number of selected anchors matching the batch size. These samples are passed through the encoder to obtain a set of latent Gaussian distributions, which serve as components of a Gaussian Mixture Model, with uniform weights assigned to all components. Since the anchor set is resampled at each training step, the prior remains adaptive and batch-dependent. This dynamic design enables the model to better capture the evolving latent structure during training, improving both stability and expressiveness compared to a fixed prior.
>
> >**Q4: What is the intuition for why the model, with a simple coordinate encoding, outperforms specialized graph neural network models on the spatial transcriptomics task?**
>
> **A4:** While specialized graph neural networks (e.g., STAGATE (NC, 2022), GraphST (NC, 2023), xSiGra (BIB, 2024)) are explicitly designed to model spatial dependencies through neighborhood graphs or attention mechanisms, they often rely on predefined graph structures or hand-crafted connectivity priors. These assumptions may not fully adapt to the irregular, noisy and highly variable nature of spatial transcriptomics data, especially at single-cell resolution. In contrast, our model leverages a simple yet effective coordinate encoding strategy (CoordConv (NeurIPS, 2018)), treating spatial coordinates as part of the input feature space. This allows the model to jointly learn spatial and transcriptomic representations in a flexible and data-driven manner, without being constrained by fixed graph topology or local smoothness assumptions. The shared latent space learns to capture global structure and local spatial patterns directly from the data, which can be more robust to technical noise and biological heterogeneity. Moreover, the strong performance of our model can be attributed to its mutually supervised framework, which unifies gene expression, spatial coordinates and morphological features. By promoting structured disentanglement and cross-modal alignment, it enables spatial structures to emerge naturally without relying on explicitly constructed spatial graphs.
>
> >**W3: The model claims to disentangle shared and private variables, but the shared latent z is inferred from a single modality before being regularized. This indirect enforcement of "sharedness" may be less effective at purging modality-specific attributes than methods that infer the shared latent from a joint posterior.**
>
> **A5:** As a matter of fact, this indirect enforcement of sharedness provides a flexible regularization which allows us to leverage mutual supervision together with mutual information-based alignment to ensure that $z$ captures cross-modal, modality-invariant information. Specifically, methods that infer the shared latent from a joint posterior can be grouped into two categories based on the adopted regularization term. For methods such as MoPoE (ICLR, 2021) and MMVAE+ (ICLR, 2023), they use the normal distribution as the regularizers for the learned joint posterior. However, this can be problematic as strong regularization would lead to less informative shared information. Moreover, in the extreme case with a disentanglement design, the modality-specific encoding with enough capacity would contain all information for the given modality and thus the shared subspace is ignored by the decoders. Alternatively, methods use VampPrior strategies such as MMVM (NeurIPS, 2024) and MEME (ICLR, 2022) fail to disentangle the shared and private latents, which inevitably results in entangled representations where modality-specific attributes may leak into the shared space, degrading the quality of the shared latent and ultimately hindering performance on downstream tasks. These limitations are also validated by our experiments: as shown in **Figures 2-3, 6-11, 25-30 and Tables 1-2**, our approach consistently outperforms these baseline methods on both the cross-modal generation and clustering tasks.

---

### Note · Authors · 2025-08-13

We thank the reviewers for their constructive feedback. We have carefully addressed the reviewers' concerns regarding model mechanism, experimental design and mathematical clarity. Following our detailed discussions, the reviewers have acknowledged that all their questions have been satisfactorily answered and unanimously agreed that **DCMEM demonstrates clear superiority in cross-modal representation learning over existing alternatives**.

Concretely, DCMEM addresses two long-standing limitations in VAE-based cross-modal frameworks: (1) **insufficient alignment of shared latent variables across modalities** and (2) **inadequate disentanglement between shared and modality-specific information**. Technically, DCMEM’s core innovations are: (1) **Structured mutual information disentanglement**—explicitly modeling shared and private latent variables, maximizing MI with observed data while minimizing redundancy between them, effectively preventing information leakage and ensuring clean factor separation. (2) **Dual-level shared latent alignment**—combining global distribution alignment via KL divergence with instance-level semantic alignment via a contrastive MI estimator, ensuring both distributional and semantic consistency of shared latents. (3) **Unified mutual supervision ELBO**—integrating disentanglement and alignment objectives into a single training formulation, enabling robust performance in generation, classification and clustering. (4) **Dynamic prior (VampPrior) mechanism**—per-batch anchor sampling that adapts the prior to the evolving latent structure, improving stability and expressiveness over fixed priors.

Our experiments on widely used multimodal benchmarks (MNIST-SVHN, CUBICC, spatial transcriptomics) show consistent improvements over strong and diverse VAE-based baselines, including challenging settings like **90% modality missingness or noisy/weak modalities**. Ablation studies confirm the necessity of each component and our rebuttal clarified methodological details, including comparison protocols, textual representation challenges and prior construction.

The reviewers recognized the method’s **technical soundness, novelty and robustness**, and we have incorporated their suggestions to enhance clarity and completeness. We believe our framework delivers both theoretical novelty and practical relevance for cross-modal representation learning and we sincerely hope that the AC will recognize the contribution and significance of this work.

---

### Decision · Program_Chairs · 2025-09-17

**Decision:**

Accept (poster)

**Comment:**

This paper presents DCMEM, a VAE-based framework for cross-modal representation learning that explicitly disentangles shared and modality-specific latent variables using mutual information regularization. The work received positive feedback from four reviewers with final ratings of 5, 4, 5, and 5, demonstrating strong agreement between reviewers. The method addresses limitations in existing cross-modal VAE approaches and shows consistent improvements across diverse tasks and challenging missing data scenarios. While some reviewers initially raised concerns about incremental novelty and technical clarity, the authors' responses and additional experiments effectively addressed these issues. The paper makes solid contributions to multimodal representation learning with clear practical applications, particularly in biomedical domains.

__Strengths__:
- Extensive evaluation across multiple datasets and tasks with particularly strong performance under severe missing data conditions
- Well-motivated combination of mutual information disentanglement, dual-level shared latent alignment, and unified mutual supervision
- Native ability to train on partially observed data without requiring separate strategies for missing modalities

__Weaknessses__:
- Core contributions largely consist of principled combinations of existing techniques rather than fundamentally new theoretical insights
- Framework specifically designed for bimodal scenarios and would require non-trivial modifications for more than two modalities
- Underperforms some baselines on certain metrics without clear explanations in the original submission